# When Will It Fail?: Anomaly to Prompt for Forecasting Future Anomalies in Time Series

**Min-Yeong Park** [* 1]   **Won-Jeong Lee** [* 2]   **Seong Tae Kim** [1]   **Gyeong-Moon Park** [2]

## Abstract

Recently, forecasting future abnormal events has emerged as an important scenario to tackle real-world necessities. However, the solution of predicting specific future time points when anomalies will occur, known as Anomaly Prediction (AP), remains under-explored. Existing methods dealing with time series data fail in AP, focusing only on immediate anomalies or failing to provide precise predictions for future anomalies. To address the AP task, we propose a novel framework called Anomaly to Prompt (A2P), comprised of Anomaly-Aware Forecasting (AAF) and Synthetic Anomaly Prompting (SAP). To enable the forecasting model to forecast abnormal time points, we adopt a strategy to learn the relationships of anomalies. For the robust detection of anomalies, our proposed SAP introduces a learnable Anomaly Prompt Pool (APP) that simulates diverse anomaly patterns using signal-adaptive prompt. Comprehensive experiments on multiple real-world datasets demonstrate the superiority of A2P over state-of-the-art methods, showcasing its ability to predict future anomalies. Our implementation code is available at https://github.com/KU-VGI/AP.

## 1. Introduction

As deep learning methods have evolved rapidly, their application to time series analysis has gained significant attention due to their critical importance in real-world scenarios. As part of this, recently, forecasting future abnormal events has been newly proposed in time series analysis, such as in

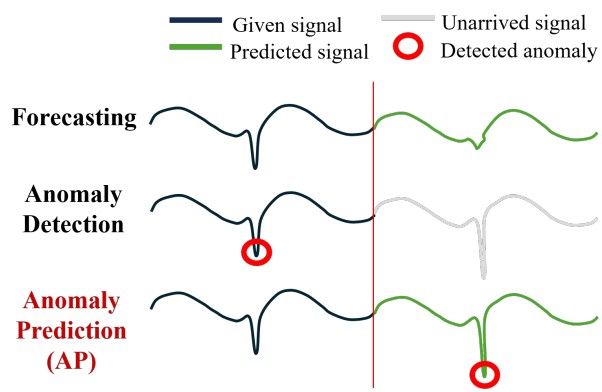

Figure 1: Comparison among different scenarios of existing time series anomaly detection, forecasting, and a newly proposed anomaly prediction.

(Jhin et al., 2023) and (You et al., 2024), aiming to enhance preparedness for potential abnormal events in real-world scenarios. For example, it is greatly helpful for medical doctors to predict potential abnormalities based on patients' biomedical data because they can make decisions about their health in advance. Another example case can be the maintenance of industrial systems where a prediction of future abnormal events is crucial, since companies or users can minimize costs from abrupt system failure.

The illustration of three different scenarios about time series forecasting, time series anomaly detection (AD), and Anomaly Prediction (AP) is depicted in Figure 1. In time series forecasting, a model should predict how future signals will look like, while in anomaly detection, a model should detect abnormal time points from the given signal. In Anomaly Prediction, a model should detect anomalies in the predicted signal. This is a more realistic and challenging scenario, in which we are interested in this paper.

Although there have been some recent attempts to define the AP scenario in which a model should predict future anomalies as in (Jhin et al., 2023) and (You et al., 2024), they cannot meet the demands of the real world. (Jhin et al., 2023) can only detect if an anomaly is likely to occur in the very near future, while failing to provide information on exact abnormal time points. In the real world, it is crucial

---
[*]Equal contribution   [1]Department of Artificial Intelligence, Kyung Hee University, Yongin, Republic of Korea [2]Department of Artificial Intelligence, Korea University, Seoul, Republic of Korea. Correspondence to: Seong Tae Kim <st.kim@khu.ac.kr>, Gyeong-Moon Park <gm-park@korea.ac.kr>.

*Proceedings of the $42^{nd}$ International Conference on Machine Learning*, Vancouver, Canada. PMLR 267, 2025. Copyright 2025 by the author(s).

to predict possible future anomalies within a more distant future, robustly against the various lengths of the future in which we are interested. However, existing works including (You et al., 2024) do not directly tackle the problems of Anomaly Prediction yet and also lack analysis on longer and various future lengths, limiting its applicability in the real world. To sum up, despite the significance of the AP scenario, how to predict the timing of abnormal events in the future is still under-explored.

A straightforward combination of existing state-of-the-art time series forecasting and anomaly detection methods may appear to be a natural baseline for the AP task, where the anomaly detection model detects anomalies from predicted signals that are the outputs of the forecasting model. Figure 2 presents a comparison of AD, which detects anomalies from past signals, and AP, which identifies anomalies from the predicted future signals using a combination of time series forecasting and anomaly detection methods. However, as shown in Figure 2, we empirically found that a naïve combination of the time series forecasting model and time series anomaly detection model fails at predicting future anomalies. The reason for this failure is quite intuitive: existing forecasting models are trained on only normal signals and predict them, thereby overlooking the prominence of abnormality in abnormal time points when predicting future signals. As a result, anomaly detection models fail at detecting anomalies because the forecasting models rather reduce the degree of abnormality of anomaly time points, which makes it difficult to detect them for anomaly detection models.

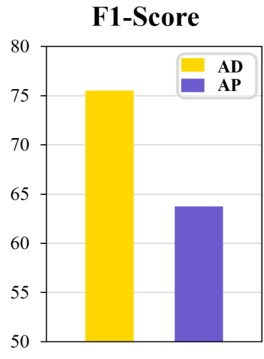

Figure 2: Comparison of F1-scores for existing time series anomaly detection task (AD) and Anomaly Prediction task (AP) in the MBA dataset.

To effectively resolve this challenging scenario, we propose a simple yet effective framework, **Anomaly to Prompt (A2P)**, which is composed of **A**nomaly-**A**ware **F**orecasting (**AAF**) and **S**ynthetic **A**nomaly **P**rompting (**SAP**). AAF aims to consider the existence of anomalies in the training process of forecasting. To achieve this, we utilize an Anomaly-Aware Forecasting Network which is pre-trained before the main training to learn the probability of being an anomaly at a specific time point. By learning the relationship of anomaly signals, we solve the absence problem of abnormality in signals while forecasting, which hinders existing forecasting models from tackling the AP task. Along with the method to endow the capability to forecast anomalous time steps, we introduce a novel SAP method with

Anomaly Prompt Pool (APP) to improve the robustness of anomaly detection. Anomaly prompts, which are learnable parameters, are utilized to intensify the diversity of signals used for reconstruction in the anomaly detection model, by capturing the characteristics of anomalies. We leverage a novel signal-adaptive prompt tuning, with a specially designed loss term to guide signals to have abnormal features and an Anomaly Prompt Pool that contains instructions for transforming normal signals into anomalous ones. Furthermore, we adopt a shared backbone architecture that can learn a unified representation, enabling forecasting and anomaly detection at once. This can remove the need for separate models, *i.e.*, enhancing efficiency, and improves the overall performance, proven by our extensive experiments.

Our main contributions can be summarized as follows:

- For the first time, we introduce a novel Anomaly to Prompt (A2P) method to address the Anomaly Prediction (AP) task, which aims to identify the time points at which abnormal events are likely to occur in the future based on the observed signals.

- We propose an unprecedented method for forecasting time points with anomalies. To achieve this, we introduce Anomaly-Aware Forecasting (AAF) to endow the forecasting ability of future signals containing anomalies.

- We propose a novel Synthetic Anomaly Prompting (SAP) method to simulate anomalies with a novel loss objective to train Anomaly Prompt Pool (APP), composed of a set of learnable parameters. This method enables our model to effectively diversify training signals through a novel signal-adaptive tuning method.

- We conducted comprehensive experiments on various real-world datasets to show the effectiveness of our proposed method and demonstrate that our method outperforms the state-of-the-art methods.

## 2. Related Work

**Time Series Forecasting.** Time series forecasting, which is the task of forecasting future signals based on historical observations, is important in terms of practicality in the real world. Previous works on time series have achieved strong prediction performance by leveraging advances in sequence modeling machine learning methods and deep neural networks such as RNN (Hochreiter & Schmidhuber, 1997; Tokgöz & Ünal, 2018; Abdel-Nasser & Mahmoud, 2019), GNN (Jiang & Luo, 2022; Wang et al., 2022b; Panagopoulos et al., 2021), and CNN (Bai et al., 2018; Livieris et al., 2020) to capture temporal dependencies. Recently, Transformers (Vaswani et al., 2017) have begun to be actively used for time series forecasting (Zhou et al., 2021; 2022;

Wu et al., 2021; Liu et al., 2021; Cirstea et al., 2022; Zhang & Yan, 2022; Nie et al., 2023; Zhou et al., 2023; Liu et al., 2024; Xu et al., 2023). However, existing forecasting models are trained on normal signals, neglecting anomalies, which leads to poor prediction of abnormal events. Additionally, these models require human interpretation for decision-making, which is time-consuming. This paper presents a novel approach to forecast abnormal events, providing direct, practical insights for decision-making.

**Time Series Anomaly Detection.** Multivariate time series anomaly detection is a crucial problem for many applications and has been widely studied. Most of the previous studies are mainly performed in an unsupervised manner considering the restriction on access to abnormal data. Contemporary works on time series anomaly detection can be divided into reconstruction-based approaches (Shen et al., 2021; 2020a; Li et al., 2019; Su et al., 2019b; Zhou et al., 2019; Yang et al., 2023; Shin et al., 2023) which find latent representations of normal time series data for reconstruction, and forecasting-based approaches (Shen et al., 2020b; Si et al., 2023). Among them, (Xu et al., 2022) proposes a new association-based method, which applies the learnable Gaussian kernel for better reconstruction. Another recent reconstruction-based model DCdetector (Yang et al., 2023) achieves a similar goal in a much more general and concise way with a dual-attention self-supervised contrastive-type structure. Existing anomaly detection models identify past anomalies, limiting real-world applicability. In this paper, we tackle a more practical challenge: predicting anomalies in future signals for the first time.

**Predicting Future Anomalies.** Recently, predicting whether anomalies are likely to occur in the future for time-series data has been studied in (Jhin et al., 2023) and (You et al., 2024). In (Jhin et al., 2023), Precursor-of-Anomaly (PoA) detection is proposed which aims to detect future anomalies in advance. However, in PoA detection, the model can only know if any anomalies will occur in the near future or not. Therefore, it cannot be used in scenarios where identifying time steps with anomalies is crucial. In response to the need to predict the specific time points of anomalies in the future, (You et al., 2024) introduces Anomaly Prediction in which a model is required to pinpoint abnormal time steps in the future. Although it formulated the AP scenario, it does not directly tackle its challenges. Related studies have been done in early accident anticipation (Liao et al., 2024; Thakur et al., 2024) for autonomous driving as well. However, they focus on detecting the possibility of an accident as early as possible within a short video clip and cannot point out when the accident will happen in the future. Thus, there is no method that can solve AP yet. In this work, we first propose a method to deal with the problems of AP.

## 3. Method

In this section, we formulate a novel scenario called Anomaly Prediction, to foresee potential anomalies in future signals in Section 3.1. Then, we explain the architecture of A2P, a unified shared backbone network to perform both forecasting and anomaly detection at once in Section 3.2. To tackle our challenging scenario effectively, in Section 3.3, we introduce a new approach called Anomaly-Aware Forecasting for more precise forecasting of abnormal time points. Furthermore, we propose a novel method coined Synthetic Anomaly Prompting which trains a newly proposed Anomaly Prompt Pool. Anomaly-Aware Forecasting Network and Anomaly Prompt Pool are jointly pre-trained in advance of the main training where the shared backbone for both forecasting and anomaly detection is trained. Finally, we summarize the total objective function in Section 3.5.

### 3.1. Scenario Description: Time Series Anomaly Prediction

Time series anomaly prediction is a scenario that aims to pinpoint the exact time steps of anomaly points in the upcoming signals. Specifically, for a given input signal $X_{in} \in \mathbb{R}^{L_{in} \times C}$, the final goal is to obtain the binary results of anomaly detection $O \in \mathbb{R}^{L_{out}}$ from the predicted signal $\hat{X}_{out} \in \mathbb{R}^{L_{out} \times C}$, where $L_{in}$ and $L_{out}$ are the lengths of the input and predicted signals, respectively, and $C$ is the number of channels in the signal. To perform anomaly prediction, we need a network for time series forecasting denoted as $\Theta_F$, and a network for time series anomaly detection denoted as $\Theta_{AD}$. Therefore, $O$ can be written as $O = \Theta_{AD} \circ \Theta_F(X_{in}) = \Theta_{AD}(\hat{X}_{out})$, where $\hat{X}_{out} = \Theta_F(X_{in})$. For the evaluation of anomaly prediction performance, F1-score is used as existing time series anomaly detection methods do. The difference in the measurement from the existing methods is that the Point Adjustment (PA) proposed in (Audibert et al., 2020) cannot be adopted in original way, since in anomaly prediction, it is important to identify specific time points. Therefore, we alleviate PA for our metric, which is explained in Section 4.

### 3.2. Unified Architecture for Anomaly Prediction

Both existing time series forecasting models such as (Nie et al., 2023; Wang et al., 2022a; Zhou et al., 2023; Wu et al., 2021; Zhou et al., 2021; 2022; Liu et al., 2021) and anomaly detection models like (Xu et al., 2022; Yang et al., 2023) capture the representation of time series data. Inspired by this point, we adopt a shared backbone $(\theta)$ to establish a unified architecture to learn the representations of signals for both the forecasting and anomaly detection models at the same time, as shown in Figure 3.

Specifically, in our framework, several base layers of transformer blocks denoted as $\theta$ are shared, while other specific parts, the embedding layers ($e_F$ and $e_{AD}$) and output layers ($o_F$ and $o_{AD}$) to construct $\Theta_F$ and $\Theta_{AD}$, exist separately, i.e., $\Theta_F = \{e_F, o_F, \theta\}$ and $\Theta_{AD} = \{e_{AD}, o_{AD}, \theta\}$. By sharing the backbone network, our model can accumulate general knowledge for both time series forecasting and anomaly detection effectively, resulting in rich representations and performance improvements. We analyze the effectiveness of the unified framework in Section 4.

### 3.3. Pre-Training of Anomaly-Aware Forecasting Network and Anomaly Prompt Pool

**Anomaly-Aware Forecasting.** We propose a novel method called Anomaly-Aware Forecasting (AAF), which improves the accuracy of future signal prediction by explicitly accounting for anomalies in prior signals. Unlike traditional forecasting models that treat all past data equally, our approach incorporates an additional module to improve robustness in dynamic and unpredictable environments. The core of this method is the Anomaly-Aware Forecasting Network, which is pre-trained to learn the complex relationships between prior signal anomalies and future trends. This pre-training step enables the network to anticipate how past anomalies might influence upcoming signals, thereby providing a more informed and accurate forecast during the main training phase.

The main purpose of AAF is to learn the relation between abnormal features inherent in a prior signal and its following future signal. To this end, we exploit Anomaly-Aware Forecasting Network which is composed of embedding layers, an attention layer, and an activation layer as shown in Figure 3. The inputs of Anomaly-Aware Forecasting Network are $X_{out}^z$ and $X_{in}^z$, which are the results of random anomaly injection among seasonal, global, trend, contextual, and shapelet anomaly types from $X_{out}$ and $X_{in}$, respectively. For the detailed implementation of the injection, we adopt the scheme used in (Darban et al., 2025). We first identify where to inject these abnormalities by comparing each signal with its reconstruction output using $f_{ftr}$, which is a pre-trained model, and focusing on the regions that yield the highest Mean Squared Error. Moreover, the magnitude of the injected abnormalities is treated as a learnable parameter, allowing the model to adaptively determine how much abnormality to inject at each identified location. The query for attention in Anomaly-Aware Forecasting Network is $e_{out}(X_{out}^z)$, which is the target that we want to know about, while key and value are $e_{in}(X_{in}^z)$, the ground for assessing the abnormality of $X_{out}^z$, where $e_{out}$ and $e_{in}$ refers to the embedding layers for $X_{out}^z$ and $X_{in}^z$. The output of the network is trained to indicate the probability of being an anomaly for each time step. This output is compared to ground truth label $y_{out}^z$ of $X_{out}^z$, and Mean Squared Error

is used for the loss term. As a result, the final loss term for training Anomaly-Aware Forecasting Network in advance of the main training is as follows:

$$\mathcal{L}_{AAF} = MSE(\sigma(Attn(e_{out}(X_{out}^z), e_{in}(X_{in}^z))), y_{out}^z), \tag{1}$$

where $\sigma$ is the activation function, sigmoid function, and $Attn$ is the cross attention layer.

**Synthetic Anomaly Prompting.** To accurately predict anomaly points, forecasting the future signal from a prior signal while considering the existence of anomaly points is crucial. To tackle this challenge, we propose a novel approach, named Synthetic Anomaly Prompting (**SAP**), which utilizes synthetic anomaly prompts for our model to predict future abnormal signals effectively. For SAP, we integrate a new Anomaly Prompt Pool (**APP**) into our unified architecture, as shown in Figure 3. The purpose of APP, which is a set of additional trainable parameters $P$, is to guide an input signal to behave like an abnormal signal, by infusing the anomaly prompts into the original embedding of the signal.

In detail, APP is defined as $P = \{(k_1, p_1), (k_2, p_2), \cdots, (k_M, p_M)\}$, where $p_m \in \mathbb{R}^{L_z \times D}$ and $k_m \in \mathbb{R}^D$ denote the $m$-th anomaly prompt and its corresponding key, respectively, $L_z$ and $D$ are the token length of single anomaly prompt and the embedding dimension, and $M$ is the pool size, which is the number of total anomaly prompts in APP. Moreover, to select $N$ best-matched prompts with the input signal $X_{in}^r$ ($X_{in}^r = \Theta_{AD}(X_{in})$) in the pool, we introduce a feature extractor $f_{ftr}(\cdot)$, which is a simple three-layer transformer architecture with a [CLS] token, as a query function, i.e., $q(X_{in}^r) = f_{ftr}([CLS; X_{in}^r])[CLS]$. The [CLS] token is a learnable embedding used to capture global representations, and it helps to select the most relevant anomaly prompt in A2P, enabling effective abnormal feature synthesis.

The process of our proposed anomaly synthesis method using APP is displayed in Figure 3. First, we pre-train the feature extractor $f_{ftr}(\cdot)$ with the train set, which will be used to select the most relevant anomaly prompt from APP. After the training of $f_{ftr}(\cdot)$, it is frozen and used for the retrieval of features from signals. Second, we train Anomaly Prompt Pool with input data $X_{in}^r$, which is the reconstructed output of $X_{in}$. The input signal passes through the feature extractor to obtain the query $q(X_{in}^r)$. This query is then matched against the keys in the APP, and the prompts corresponding to the top-$N$ closest keys are attached to the embedded input $\widetilde{X}_{in}^r \in \mathbb{R}^{L_{in} \times D}$, where $\widetilde{X}_{in}^r = e_{AD}(X_{in}^r)$. Note that the synthesis of anomaly is executed at the embedding level, which enables more diverse prompting in high dimensions. Finally, the simulated embedding of anomaly $\widetilde{X}_{in}^p$ is defined as follows:

$$\widetilde{X}_{in}^p = [p_{s_1}; \cdots; p_{s_N}; \widetilde{X}_{in}^r], \quad s_i \in \mathbf{S}, \tag{2}$$

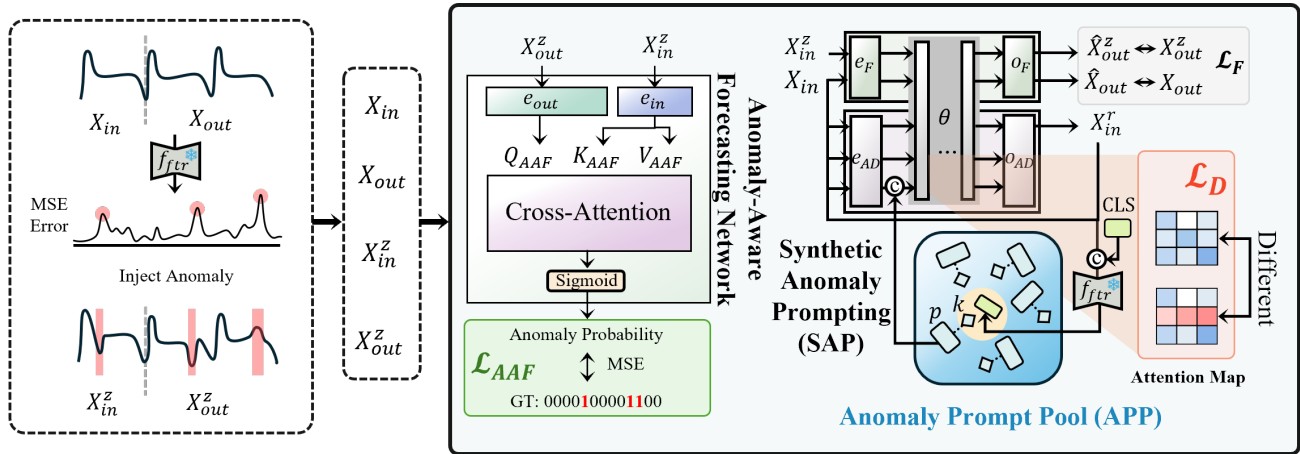

Figure 3: **Pre-training of Anomaly-Aware Forecasting Network and Anomaly Prompt Pool.** Our model first pre-trains Anomaly-Aware Forecasting Network and Anomaly Prompt Pool (APP) by injecting anomalies to train data. After pre-training, Anomaly-Aware Forecasting Network and APP are frozen in the main training.

$$\mathbf{S} = \underset{\{s_i\}_{i=1}^N \subseteq [1,M]}{\mathrm{argmax}} \sum_{i=1}^{N} \gamma \left(q(X_{in}^r), k_{s_i}\right), \qquad (3)$$

where the score function $\gamma$ is the cosine similarity, which is for calculating how each anomaly prompt is related to each normal signal, and $[\cdot;\cdot]$ denotes the concatenation. The selected prompt tokens are attached to the input tokens of $\widetilde{X}_{in}^r$, after passing through the embedding layer $e_{AD}$, to transform the original normal signal into an abnormal signal. The output anomaly prompts are then removed before the final projection of each $o_F$ and $o_{AD}$, to match the dimension.

**Divergence Loss.** To make the model detect more diverse anomalies, we pre-train the Anomaly Prompt Pool which holds the knowledge of characteristics of anomalies. The pre-trained APP can then be used to infuse plausible anomalies later in the main training phase. For the efficient pre-training of APP, we introduce a novel Divergence loss ($\mathcal{L}_D$) to guide the anomaly prompts in the APP to prompt the signals to function as anomalies, distinct from normal signals. Along with a term to make abnormal signals, we add an additional term to pull the selected keys closer to the corresponding features of normal signals as follows:

$$\mathcal{L}_D = -KL(A(\widetilde{X}_{in}^p), A(\widetilde{X}_{in}^r)) - \lambda_k\, \gamma\left(f_{ftr}(X_{in}^r), k_m\right). \tag{4}$$

Here, we obtain the reconstruction output $X_{in}^r$ which plays a role of pseudo-normal signal, where $A$ is the first attention layer in $\theta$. Then, the model attaches anomaly prompts from Anomaly Prompt Pool to pseudo-normal embedding $\widetilde{X}_{in}^r$ to simulate anomaly, which results in $\widetilde{X}_{in}^p$. Since we aim to train APP to add abnormalities into the signal, the features of synthetic anomaly and pseudo-normal input features are trained to be distinct with $\mathcal{L}_D$, which serves to intensify the gap between the features of pseudo-normal feature and

synthetic anomaly feature. Note that after the pre-training phase, the proposed Anomaly-Aware Forecasting Network and APP are frozen.

**Forecasting Loss.** Along with Anomaly-Aware Forecasting Network and APP, the forecasting model is pre-trained to predict future time series with anomaly signals, as well as normal signals. The outputs of $X_{in}$ and $X_{in}^z$ from the forecasting model $\Theta_F$, which result in $\hat{X}_{out}$ and $\hat{X}_{out}^z$, respectively, are used to pre-train $\Theta_F$ via forecasting loss $\mathcal{L}_F$ as follows:

$$\mathcal{L}_F = \frac{1}{2}\left(\left\|\hat{X}_{out} - X_{out}\right\|^2 + \left\|\hat{X}_{out}^z - X_{out}^z\right\|^2\right). \tag{5}$$

### 3.4. Main Training

In the main training stage, the pre-trained Anomaly-Aware Forecasting Network, explained in section 3.3, is used to output anomaly probability, as indicated in Figure 4. Therefore, in the main training, the final loss term regarding forecasting is as follows:

$$\mathcal{L}_{AF} = g(X_{in}, \hat{X}_{out}) \odot \left\|\hat{X}_{out} - X_{out}\right\|^2, \tag{6}$$

where $g(\cdot)$ is Anomaly-Aware Forecasting Network, $\hat{X}_{out}$ is $\Theta_F(X_{in})$ and $\odot$ is element-wise multiplication. By considering the errors in anomaly time steps more than other time steps, the network can be trained to focus on abnormal areas.

We employ an additional loss term that aims to reconstruct $\widetilde{X}_{in}^p$ to its original normal form $X_{in}$, as follows:

$$\mathcal{L}_R = \frac{1}{2}\left(\|X_{in} - X_{in}^{p,r}\|^2 + \|X_{in} - X_{in}^r\|^2\right), \tag{7}$$

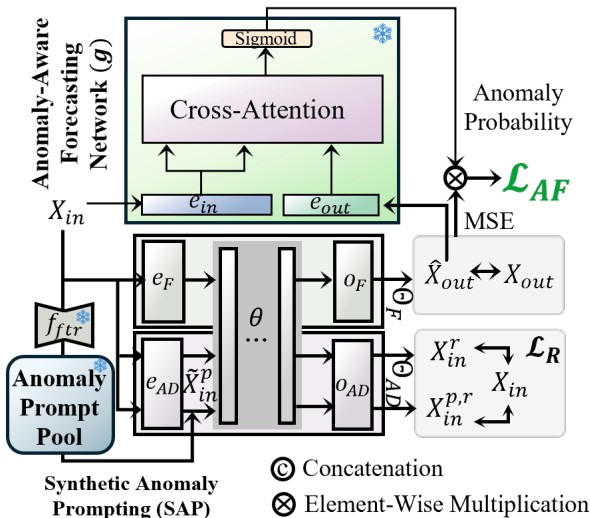

Figure 4: **Main Training of A2P.** Only the shared backbone is trained during the main training, and others are frozen.

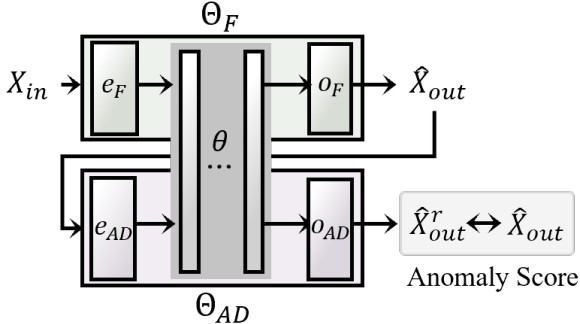

Figure 5: **Test time of Anomaly Prediction.**

where $X_{in}^{p,r} = o_{AD}(\theta(\widetilde{X}_{in}^p))$ is the reconstruction output of synthesized abnormal input embedding and $X_{in}^r = \Theta_{AD}(X_{in})$ is that of original normal input signal.

### 3.5. Total Objective Function

The total objective function of our proposed framework is summarized as follows:

$$
\mathcal{L}_{Total} = \begin{cases} \lambda_{AAF}\mathcal{L}_{AAF} + \lambda_D \mathcal{L}_D + \lambda_F \mathcal{L}_F & \text{for PT,} \\ \lambda_R \mathcal{L}_R + \lambda_{AF}\mathcal{L}_{AF}, & \text{for MT,} \end{cases}
$$
(8)

where PT and MT mean pre-training and main training, respectively, and $\lambda = \{\lambda_{AAF}, \lambda_D, \lambda_F, \lambda_R, \lambda_{AF}\}$ is a set of coefficients for weighting each loss term. We set all five coefficients to 1 as default values during whole experiments.

### 3.6. Test Time

In the test time for the evaluation of A2P, only the shared backbone is utilized and other modules are discarded as

shown in Figure 5. First, the forecasted signal $\hat{X}_{out}$ is obtained by forwarding the input signal $X_{in}$ to $\Theta_F$. Then the reconstructed output of $\hat{X}_{out}$ is obtained as $\hat{X}_{out}^r = \Theta_F(\hat{X}_{out})$. Finally, the anomaly score for each time step is calculated using $\hat{X}_{out}$ and $\hat{X}_{out}^r$, following the scheme of (Xu et al., 2022).

## 4. Experiments

### 4.1. Experimental Setup

**Dataset Configurations.** We evaluated our method on four real-world time series datasets: 1) MBA (MIT-BIH Supraventricular Arrhythmia Database) (Moody & Mark, 2001) is a set of electrocardiogram recordings from four patients, composed of two distinct types of irregularities (supraventricular contractions or premature heartbeats). 2) Exathlon (Jacob et al., 2020) is a set of real-world datasets collected using Apache Spark. It is comprised of eight sub-datasets, each dataset with 19 dimensions. 3) SMD (Server Machine Dataset) (Su et al., 2019a) is a 5-week-long dataset that was collected from a large Internet company with 38 dimensions. 4) WADI (Water Distribution) (Ahmed et al., 2017) is a distribution system comprising a larger number of water distribution pipelines with 123 dimensions.

**Baselines and Evaluation Metrics.** We compared our model with various combinations of existing forecasting models and anomaly detection models, considering them as our baselines. For forecasting models, we adopted state-of-the-art models, PatchTST (Nie et al., 2023), MICN (Wang et al., 2022a), GPT2 (Zhou et al., 2023), iTransformer (Liu et al., 2024), and FITS (Xu et al., 2023). Regarding anomaly detection models, we adopted reconstruction-based methods, AnomalyTransformer (Xu et al., 2022), DCDetector (Yang et al., 2023), and CAD (Si et al., 2023). We conducted additional baseline experiments, which are described in Appendix C.3. We used F1-score (F1) as the main evaluation metric. If not mentioned, the scores reported in the tables indicate F1-scores. In addition, F1-score was calculated without point adjustment introduced in (Audibert et al., 2020). Instead, we used F1-score with tolerance $t$, which denotes the time window within which errors are tolerated in anomaly detection. For example, when a model predicts that a time step $i$ is an anomaly, the real ground-truth anomaly time points from $[i - t, i + t]$ are considered to be correctly detected before the calculation of F1-score.

**Hyperparameters.** The hyperparameters used in our experiments are mentioned in Appendix A.2, and the sensitivity results on various parameter values are shown in Appendix E.

Table 1: Anomaly Prediction results on multivariate cases with $L_{in} = 100$, averaged over 3 random seeds. The **best** and second-best results are highlighted.

| $L_{out}$ | Model | | Dataset | | | | Avg. F1 |
| | F | AD | MBA | Exathlon | SMD | WADI | |
|---|---|---|---|---|---|---|---|
| 100 | P-TST | AT | 55.05±3.75 | 18.10±0.24 | 34.32±0.50 | 58.02±3.95 | 41.37 |
| | | DC | 59.59±3.94 | 17.21±0.17 | 23.01±5.31 | 18.50±13.99 | 29.08 |
| | | CAD | 53.75±0.11 | 18.10±0.36 | 25.55±0.10 | 54.33±2.10 | 37.93 |
| | MICN | AT | 38.84±14.85 | 17.30±1.29 | 34.06±1.38 | 54.51±3.99 | 36.18 |
| | | DC | 57.90±1.95 | 17.35±1.19 | 31.80±1.63 | 10.21±3.23 | 29.82 |
| | | CAD | 57.42±0.52 | 5.19±0.44 | 18.17±1.90 | 18.45±7.27 | 24.81 |
| | GPT2 | AT | 49.20±7.02 | 17.83±3.20 | 29.49±2.46 | 59.88±7.03 | 39.60 |
| | | DC | 55.13±3.80 | 9.66±0.74 | 31.23±1.98 | 21.56±7.50 | 29.90 |
| | | CAD | 16.64±8.75 | 17.34±1.13 | 8.14±0.57 | 26.97±2.53 | 17.77 |
| | iTransformer | AT | 54.72±5.35 | 17.82±0.81 | 32.75±0.80 | 54.54±8.28 | 39.96 |
| | | DC | 53.07±5.30 | 12.98±3.73 | 13.47±6.62 | 13.65±12.82 | 23.29 |
| | | CAD | 39.76±6.38 | 16.94±0.88 | 25.69±2.58 | 24.36±1.60 | 26.69 |
| | FITS | AT | 41.51±5.33 | 17.74±2.85 | 32.65±1.39 | 60.30±4.46 | 39.55 |
| | | DC | 61.39±4.89 | 17.38±3.26 | 32.45±1.27 | 52.96±6.11 | 41.55 |
| | | CAD | 23.72±9.43 | 10.87±0.06 | 33.15±1.14 | 28.93±0.45 | 24.67 |
| | **A2P (Ours)** | | **67.55±5.62** | **18.64±0.16** | **36.29±0.18** | **64.91±0.47** | **46.84** |
| 200 | P-TST | AT | 51.25±3.52 | 17.65±0.31 | 34.07±1.88 | 55.24±6.62 | 39.55 |
| | | DC | 59.04±3.69 | 16.23±0.50 | 13.42±5.38 | 7.36±5.20 | 24.01 |
| | | CAD | 16.87±5.21 | 16.28±0.11 | 6.71±0.79 | 31.69±1.41 | 17.89 |
| | MICN | AT | 56.57±2.90 | 17.30±0.09 | 33.44±1.81 | 55.24±3.87 | 40.64 |
| | | DC | 57.80±1.32 | 17.41±0.34 | 30.32±2.13 | 56.24±1.13 | 40.94 |
| | | CAD | 50.46±0.82 | 2.44±0.07 | 6.36±0.23 | 32.75±2.00 | 23.00 |
| | GPT2 | AT | 49.29±2.55 | 17.64±0.54 | 35.12±0.90 | 59.35±4.34 | 40.35 |
| | | DC | 55.44±2.84 | 10.25±0.12 | 9.77±4.67 | 26.14±3.10 | 25.09 |
| | | CAD | 26.24±1.78 | 15.40±0.13 | 7.42±0.37 | 27.87±2.11 | 19.73 |
| | iTransformer | AT | 54.13±0.08 | 17.88±0.14 | 27.26±6.01 | 48.72±9.47 | 37.50 |
| | | DC | 52.53±7.47 | 14.86±0.05 | 3.65±4.36 | 14.45±20.44 | 21.87 |
| | | CAD | 21.14±7.35 | 16.58±0.08 | 7.66±0.23 | 30.37±2.39 | 18.44 |
| | FITS | AT | 47.85±2.07 | 17.88±0.26 | 35.70±0.24 | 64.10±1.22 | 41.38 |
| | | DC | 51.21±7.64 | 17.56±0.21 | 29.69±1.57 | 53.60±3.49 | 38.02 |
| | | CAD | 51.44±14.45 | 10.04±0.41 | 9.32±0.70 | 32.75±2.00 | 25.39 |
| | **A2P (Ours)** | | **74.63±5.92** | **28.71±0.54** | **42.36±0.80** | **66.65±1.93** | **53.08** |
| 400 | P-TST | AT | 47.25±11.15 | 17.30±0.49 | 26.56±1.50 | 56.05±1.34 | 36.79 |
| | | DC | 56.89±8.21 | 16.65±0.32 | 14.55±8.78 | 47.36±1.51 | 33.36 |
| | | CAD | 20.04±9.04 | 13.40±0.31 | 3.75±0.20 | 20.22±0.78 | 14.35 |
| | MICN | AT | 57.81±1.57 | 17.37±0.75 | 35.71±0.10 | 53.83±1.96 | 41.18 |
| | | DC | 56.08±5.30 | 17.49±0.49 | 29.95±3.63 | 57.80±3.88 | 40.83 |
| | | CAD | 21.12±1.87 | 13.40±0.10 | 3.86±0.10 | 5.02±0.26 | 10.85 |
| | GPT2 | AT | 50.31±5.40 | 17.53±0.89 | 32.89±3.49 | 52.86±6.47 | 38.90 |
| | | DC | 57.70±10.28 | 9.49±3.29 | 3.89±3.06 | 15.16±1.80 | 21.06 |
| | | CAD | 3.57±2.25 | 11.07±4.22 | 4.14±0.72 | 11.07±4.22 | 7.96 |
| | iTransformer | AT | 49.14±3.74 | 17.41±0.70 | 32.89±3.49 | 53.17±5.58 | 38.15 |
| | | DC | 49.72±4.23 | 15.46±0.22 | 12.61±7.20 | 7.57±10.70 | 21.84 |
| | | CAD | 24.87±6.02 | 12.37±1.13 | 3.55±0.83 | 10.23±8.44 | 12.76 |
| | FITS | AT | 46.60±5.77 | 18.03±0.95 | 33.49±2.99 | 56.66±5.89 | 38.20 |
| | | DC | 54.32±11.29 | 17.66±0.42 | 23.83±11.40 | 45.12±1.78 | 35.73 |
| | | CAD | 66.33±4.21 | 7.50±0.56 | 4.95±0.12 | 22.60±0.08 | 25.84 |
| | **A2P (Ours)** | | **69.35±7.15** | **43.57±1.10** | **48.10±2.55** | **74.57±6.37** | **58.89** |

**Anomaly Threshold.** The threshold for deciding anomalies from anomaly scores is set by following the widely accepted protocol from (Shen et al., 2020a), adjusting for a percentage of anomalies in the test data. This approach ensures consistency with established standards for anomaly detection tasks.

## 4.2. Anomaly Prediction Results

The results of the Anomaly Prediction experiments are demonstrated in Table 1. For the F1-score, our model consistently outperforms the baselines, showing the effectiveness of our proposed Anomaly-Aware Forecasting and Synthetic Anomaly Prompting. Note that ours were effective in datasets from various domains, which implies that our methods are robust to the various statistics of datasets.

## 4.3. Ablation Study

**AAF and SAP.** To further examine the effectiveness of our novel methods, we thoroughly conducted ablation studies. The ablation results of AAF and SAP are indicated in Table 2 to see the impact of each method. As shown in Table 2, 1) utilizing the knowledge of the relationship among anomaly signals and 2) synthesizing anomalies at the embedding level in a learnable way improved Anomaly Prediction performance, respectively.

**Pre-training Loss Ablation.** Table 3 demonstrates the ablation of two loss terms used in the pre-training phase. For training the backbone, utilizing $\mathcal{L}_F$ and our proposed $\mathcal{L}_D$ both contributed to significant performance enhancement. Especially, the Divergence loss ($\mathcal{L}_D$) alone improved about 24% of its performance in the MBA dataset, emphasizing the effectiveness of $\mathcal{L}_D$ for the robust reconstruction of time series data from various abnormal features.

**Shared Backbone.** In order to investigate the impact of sharing transformer layers between the forecasting model and the anomaly detection model, we conducted ablation experiments regarding the effectiveness of the shared backbone as demonstrated in Table 4. Sharing the layers of backbone for forecasting and anomaly detection remarkably enhanced the performances, implying that sharing the knowledge of forecasting and anomaly detection helped to enrich the representation learning of time series signals.

**Using Anomaly Probability for AAF.** To investigate the effectiveness of using anomaly probability for training the forecasting model in main training phase, we ablated the use of anomaly probability in Table 5. Using weighted loss term $\mathcal{L}_{AF}$ notably enhanced the performance of AP compared to using traditional forecasting loss, indicating considering anomaly probability in forecasting process is effective.

Table 2: Ablation for the proposed AAF and SAP, when $L_{in} = L_{out} = 100$.

| AAF | SAP | MBA | Exathlon | SMD | WADI | Avg. F1 |
|---|---|---|---|---|---|---|
| ✗ | ✗ | 36.26 | 17.65 | 34.74 | 58.66 | 36.82 |
| ✓ | ✗ | 40.95 | 17.76 | 34.87 | 61.98 | 38.89 |
| ✗ | ✓ | 55.95 | 18.29 | 36.05 | 59.36 | 42.41 |
| ✓ | ✓ | **67.55** | **18.64** | **36.29** | **64.91** | **46.84** |

Table 3: Ablation of loss terms $\mathcal{L}_D$ and $\mathcal{L}_F$ used in pre-training.

| $\mathcal{L}_F$ | $\mathcal{L}_D$ | MBA | Exathlon | SMD | WADI | Avg. F1 |
|---|---|---|---|---|---|---|
| ✗ | ✗ | 40.95 | 17.76 | 34.87 | 61.98 | 38.89 |
| ✓ | ✗ | 50.57 | 17.91 | 34.92 | 62.31 | 41.42 |
| ✗ | ✓ | 65.19 | 17.91 | 35.17 | 63.25 | 45.38 |
| ✓ | ✓ | **67.55** | **18.64** | **36.29** | **64.91** | **46.84** |

Table 4: Ablation for the shared transformer backbone.

| Shared | MBA | Exathlon | SMD | WADI | Avg. F1 |
|---|---|---|---|---|---|
| ✗ | 51.53 | 18.00 | 35.60 | 60.70 | 41.45 |
| ✓ | **67.55** | **18.64** | **36.29** | **64.91** | **46.84** |

Table 5: Ablation for the use of anomaly probability in main training.

| Forecasting Loss | MBA | Exathlon | SMD | WADI | Avg. F1 |
|---|---|---|---|---|---|
| MSE | 64.20 | 18.24 | 36.13 | 59.19 | 44.44 |
| MSE $\otimes$ AN. Prob. | **67.55** | **18.64** | **36.29** | **64.91** | **46.84** |

Table 6: Forecasting performances evaluated by MSE of forecasting models and A2P (Ours) on the MBA dataset.

| $L_{out}$ | 100 | 200 | 400 |
|---|---|---|---|
| PatchTST | 1.174 | 1.261 | 1.272 |
| MICN | 1.012 | 1.017 | 1.041 |
| GPT2 | 1.021 | 1.096 | 1.100 |
| iTransformer | 1.140 | 1.256 | 1.279 |
| FITS | 1.495 | 1.844 | 2.390 |
| **A2P (Ours)** | **0.788** | **0.864** | **0.930** |

## 4.4. Analysis

**Results on Forecasting.** As shown in Table 6, our proposed A2P was advantageous at predicting more accurate future signals. The performance improvements in not only Anomaly Prediction but also forecasting indicate that our proposed approaches contributed to learning the representations of both normal and abnormal signals effectively, compared to the baseline. Notably, when $L_{out}$ was significantly longer, our proposed A2P outperformed at forecasting future signals with anomalies, indicating the capability of handling long-term signals.

**Qualitative Results on Anomaly Prediction.** To further examine the effectiveness of our method, we visualized the time series signals as shown in Figure 6. As shown in the figure, our proposed A2P successfully forecasted the signal whereas our baseline, the naive combination of PatchTST and AnomalyTransformer failed. Our proposed

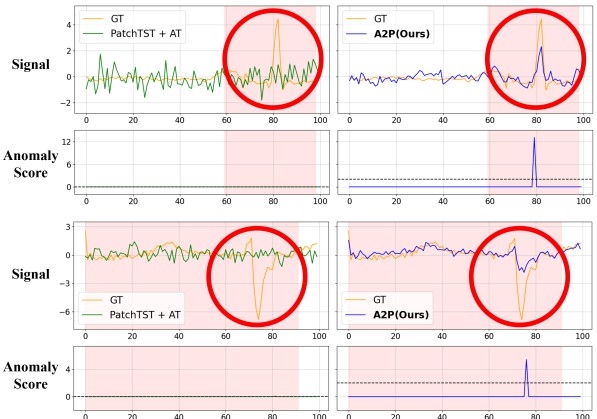

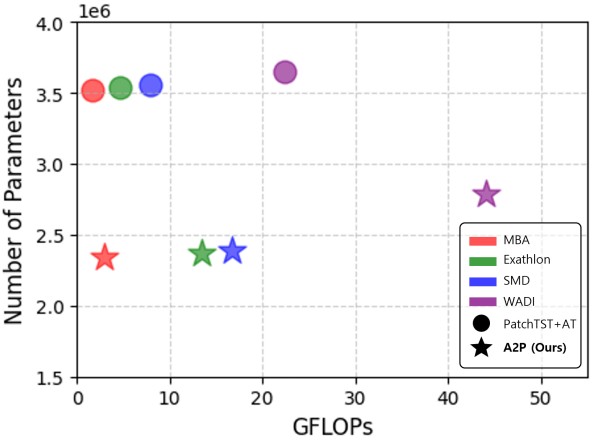

Figure 6: Ground-truth and predicted signals of MBA dataset from the baseline and A2P (top), with corresponding anomaly scores (bottom). The black dotted line represents the threshold of anomaly scores, and red-shaded area indicates time step with anomalies.

Figure 7: Comparison of GFLOPs and the number of parameters.

A2P predicted the abnormal events appropriately, and it led to successful anomaly detection.

**Computational Complexity.** To investigate the impact of additional computational complexity of our proposed methods, we measured GFLOPs per iteration (including both pre-training and main training) and the total number of parameters of the baseline (combination of PatchTST and AnomalyTransformer) and our proposed A2P. As shown in the Figure 7, A2P does require additional computation compared to the baseline. However, since our model unifies both forecasting and anomaly detection within a single framework, it significantly reduces the overall parameter footprint. More importantly, this additional cost is only incurred during training, with no extra overhead at inference time. Despite the modest increase in training complexity,

A2P achieved an average 10% improvement in performance over the baseline, demonstrating a favorable trade-off between cost and accuracy. Moreover, the train time consumed is utmost 1 hour in WADI dataset, which is negligible and is not really a heavy burden for A2P to be applied in real-world scenarios.

## 5. Conclusion

In this paper, we first addressed a solution to Anomaly Prediction (AP), where the model needs to detect abnormal time points from unarrived future signals. We tackle AP by employing synthetic anomalies in train time, whereas traditional time series forecasting and anomaly detection models were trained with only normal signals, limiting their generalizability to abnormal signals. We proposed two effective approaches, Anomaly-Aware Forecasting (AAF) and Synthetic Anomaly Prompting (SAP). In AAF, we designed Anomaly-Aware Forecasting Network to help the model forecast time steps with anomalies. In SAP, we defined APP which learns how to prompt the input signals to have anomalous features. We achieved state-of-the-art performances on the AP task in various real-world datasets, demonstrating the effectiveness of our methods through comprehensive experiments. We hope our pioneering attempt to predict future anomalies provides an opportunity to anticipate potential breakdowns, while also opening up a new direction for research.

## Acknowledgement

This work was supported by Korea Planning & Evaluation Institute of Industrial Technology (KEIT) grant funded by the Korea government (MOTIE) (RS-2024-00444344), and in part by Institute of Information & communications Technology Planning & Evaluation (IITP) grant funded by the Korea government (MSIT) (No. RS-2019-II190079, Artificial Intelligence Graduate School Program (Korea University), and No. RS-2024-00457882, AI Research Hub Project).

## Impact Statement

This paper presents research aimed for the advance of Machine Learning by contributing novel methodologies for predicting future anomalies from unarrived time series data. Our work has the potential to influence various domains which utilize time series data, including artificial intelligence, data science, and automated decision-making. While we recognize that any progress in time series analysis may have broad societal implications—ranging from ethical considerations to real-world applications in industry and academia—we do not identify any specific consequences that warrant particular emphasis at this time. However, we encourage further discussion and evaluation of the broader impact as the field continues to evolve.

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

# A. Additional Information

## A.1. Notations

Table 7: Categorized notations.

| Symbol | | Description |
|---|---|---|
| | $L_{in}$ | Length of input signal |
| | $L_{out}$ | Length of forecasted signal |
| | C | Dimension of time series data |
| Dimension | $M$ | Number of anomaly prompts in Anomaly Prompt Pool |
| | $N$ | Number of prompts to attach in Synthetic Anomaly Prompting |
| | $L_z$ | Length of single token of an anomaly prompt |
| | $D$ | Embedding dimension |
| | $\theta$ | Shared transformer blocks |
| | $e_F$ | Embedding layer for forecasting |
| | $e_{AD}$ | Embedding layer for anomaly detection |
| | $o_F$ | Projection layer for forecasting |
| Parameters | $o_{AD}$ | Projection layer for reconstruction |
| | $\Theta_F$ | Forecasting network |
| | $\Theta_{AD}$ | Anomaly detection network |
| | $f_{ftr}$ | Feature extractor for obtaining class token |
| | $P$ | Set of parameters of Anomaly Prompt Pool |
| | $X_{in}$ | Ground truth prior time series |
| | $X_{out}$ | Ground truth posterior time series |
| | $X_{in}^z$ | Time series after injecting random anomaly to $X_{in}$ |
| | $X_{out}^z$ | Time series after injecting random anomaly to $X_{out}$ |
| Signal | $\hat{X}_{out}$ | Forecasted time series from $X_{in}$ |
| | $\hat{X}_{out}^z$ | Forecasted time series from $X_{in}^z$ |
| | $X_{in}^r$ | Reconstructed output of $X_{in}$ |
| | $X_{in}^{p,r}$ | Reconstructed output of $\tilde{X}_{in}^p$ |
| | $\hat{X}_{out}^r$ | Reconstructed output of $\hat{X}_{out}$ |
| | $\tilde{X}_{in}$ | Embedding feature of $X_{in}$ |
| Feature | $\tilde{X}_{in}^p$ | Embedding feature of prompted anomaly |
| | $\tilde{X}_{in}^r$ | Embedding feature of $X_{in}^r$ |
| Etc. | $y_{out}^z$ | Ground truth anomaly label of $X_{out}^z$ |

## A.2. Training Details

For all models, we trained all models using Standard scaler, Adam optimizer (Kingma & Ba, 2015) with $\beta_1 = 0.9$ and $\beta_2 = 0.999$, a batch size of 16, and a constant learning rate of 0.0001 for all settings. We used the length of input sequence $L_{in}$ as 100 for all settings. Also, we conducted experiments varying the length of output sequence $L_{out}$ from 100 to 400. For Synthetic Anomaly Prompting, we adopted the length of anomaly prompts $L_z$ and the size of anomaly prompt pool 5 and 10, respectively, as default. In the selection of top-$N$ anomaly prompts in anomaly prompt pool, we used $N = 3$. We trained the models for 5 epochs, with 3 layers of transformer backbone, and the embedding dimension $D$ is fixed to 256. Also, our experiments were executed on single GPU (NVIDIA RTX 3090), implementation library (PyTorch (Paszke et al., 2019)) for fair and exhaustive comparison. Regarding the anomaly detection model for Anomaly Prediction, we set the window size of 100, and used sliced predicted signals to obtain the output of anomaly detection in experiments, including all comparing methods.

## A.3. Dataset Details of the Anomalies

Table 8 provides detailed information about the anomaly configurations of the datasets.

Table 8: Statistics of Anomalies in the Dataset.

| Dataset | Avg. Anomaly Ratio (%) | Avg. Anomaly Len. | # of Anomaly Seg. | # of Batches | # of Anomaly Seg. / # of Total Sample | # of Anomaly Seg. / # of Anomaly Sample |
|---------|------|------|------|------|------|------|
| MBA | 33.80 | 29.48 | 86 | 75 | 1.14 | 1.18 |
| Exathlon | 12.69 | 91.94 | 1091 | 8911 | 0.12 | 1.00 |
| SMD | 4.15 | 47.26 | 623 | 7083 | 0.08 | 1.00 |
| WADI | 4.88 | 62.19 | 27 | 344 | 0.07 | 1.00 |

Table 9: Ablation for the type of loss used in Anomaly-Aware Forecasting Network.

| Type | MBA | Exathlon | SMD | WADI | Avg. F1 |
|------|-----|----------|-----|------|---------|
| BCE | 66.75 | 17.91 | 34.82 | 64.21 | 45.92 |
| MSE | **67.55** | **18.64** | **36.29** | **64.91** | **46.84** |

Table 10: Anomaly Prediction results on datasets that are excluded in the main paper with $L_{in} = L_{out} = 100$, averaged over 3 random seeds. The **best** and second-best results are highlighted.

| Model | | Dataset | | | | Avg. F1 |
|-------|-----|------|------|------|------|---------|
| F | AD | MSL | PSM | SWaT | SMAP | |
| P-TST | AT | 41.96±0.97 | 13.74±0.52 | 11.29±0.17 | 14.97±0.16 | 20.49 |
| | DC | 39.93±1.20 | 13.88±0.30 | 7.27±2.09 | 13.91±0.09 | 18.75 |
| | CAD | 17.99±0.25 | 3.26±0.04 | **16.65±0.18** | 3.76±0.10 | 10.42 |
| MICN | AT | 39.91±0.39 | 14.04±0.25 | 11.68±0.31 | 14.74±0.67 | 20.09 |
| | DC | 41.05±0.51 | 13.87±0.51 | 11.85±0.33 | 15.14±0.93 | 20.47 |
| | CAD | 3.82±0.05 | 2.75±0.00 | 6.13±1.39 | 1.62±0.03 | 3.58 |
| GPT2 | AT | 41.91±2.73 | 13.89±0.46 | 11.18±0.46 | 15.34±0.73 | 20.58 |
| | DC | 39.31±1.72 | 10.31±5.67 | 11.00±1.15 | 9.89±1.54 | 17.63 |
| | CAD | 2.08±0.29 | 3.65±0.31 | 6.76±0.91 | 9.17±0.23 | 5.41 |
| iTransformer | AT | 42.35±0.75 | 13.83±0.96 | 11.18±0.46 | 15.87±0.24 | 20.82 |
| | DC | 39.95±0.87 | 13.46±0.41 | 4.92±2.75 | 15.54±0.35 | 18.47 |
| | CAD | 4.49±0.15 | 3.80±0.40 | 13.10±1.51 | 3.71±0.55 | 6.28 |
| FITS | AT | 41.97±0.12 | 13.79±0.32 | 11.34±0.49 | 15.35±0.55 | 20.60 |
| | DC | 41.72±1.39 | 13.65±0.61 | 9.24±1.68 | 14.21±0.34 | 19.70 |
| | CAD | 7.56±0.34 | 7.87±0.40 | 16.47±0.02 | 4.67±0.27 | 9.14 |
| **A2P (Ours)** | | **46.87±0.36** | **15.28±0.09** | 15.74±0.35 | **16.31±0.12** | **23.55** |

## B. Loss Ablation of Anomaly-Aware Forecasting Network

We also ablated the loss term used for training Anomaly-Aware Forecasting Network, as indicated in Table 9. When Binary Cross Entropy loss was used, the performance was suboptimal. The result implies that driving anomaly probability to be continuous (MSE) rather than discrete (BCE) is better to learn Anomaly-Aware Forecasting Network effectively. For this reason, we finally adopt MSE instead of BCE to learn anomaly probability in AAF.

## C. Quantitative Results

### C.1. Experiment on other datasets

Aside from the datasets that were included in the main paper, there are datasets mainly used for the evaluation of time series anomaly detection such as MSL, SMAP (Hundman et al., 2018), PSM (Abdulaal et al., 2021), and SWaT (Mathur & Tippenhauer, 2016). However, they are not appropriate for the evaluation of anomaly detection task due to the soundness

Table 11: Anomaly Prediction results evaluated using VUS-PR and VUS-ROC with $L_{in} = L_{out} = 100$, averaged over 3 random seeds. The **best** and second-best results are highlighted.

| Model | | $L_{out}$ | | | | | | Avg. | |
| --- | --- | --- | --- | --- | --- | --- | --- | --- | --- |
| | | 100 | | 200 | | 400 | | | |
| F | AD | V-PR | V-ROC | V-PR | V-ROC | V-PR | V-ROC | V-PR | V-ROC |
| P-TST | AT | 68.28 | 71.47 | 67.87 | 72.86 | 66.99 | 79.54 | 67.71 | 74.62 |
| | DC | 68.52 | 72.11 | 67.44 | 71.31 | 67.84 | 70.25 | 67.93 | 71.22 |
| | CAD | 55.26 | 68.07 | 57.98 | 71.94 | 56.96 | 70.55 | 56.73 | 70.19 |
| MICN | AT | 68.20 | 71.47 | 67.80 | 78.79 | 67.09 | 80.55 | 67.70 | 76.94 |
| | DC | 68.55 | 72.31 | 67.89 | 70.93 | 67.96 | 71.44 | 68.13 | 71.56 |
| | CAD | 67.30 | 78.31 | 67.93 | 80.12 | 60.48 | 73.36 | 65.24 | 77.26 |
| GPT2 | AT | 68.24 | 71.85 | 67.61 | 76.61 | 66.95 | 75.70 | 67.60 | 74.72 |
| | DC | 68.37 | 75.16 | 67.82 | 71.95 | 68.50 | 74.18 | 68.23 | 73.76 |
| | CAD | 57.73 | 68.41 | 55.58 | 69.02 | 55.08 | 65.85 | 56.13 | 67.76 |
| iTransformer | AT | 67.95 | 72.87 | 66.90 | 80.38 | 66.72 | 80.04 | 67.19 | 77.76 |
| | DC | 68.11 | 71.72 | 67.67 | 70.49 | 67.83 | 70.61 | 67.87 | 70.94 |
| | CAD | 66.94 | 77.10 | 66.21 | 76.91 | 67.10 | 75.62 | 66.75 | 76.54 |
| FITS | AT | 67.98 | 71.59 | 68.14 | 72.30 | 68.24 | 72.57 | 68.12 | 72.15 |
| | DC | 68.54 | 71.82 | 67.79 | 70.56 | 67.98 | 71.25 | 68.10 | 71.21 |
| | CAD | 68.03 | 80.37 | 67.92 | 80.70 | 67.43 | 80.30 | 67.79 | 80.46 |
| **A2P (Ours)** | | **71.18** | **83.37** | **68.17** | **81.55** | **69.01** | **82.52** | **69.46** | **82.48** |

of the datasets as discussed in (Wagner et al., 2023). Though, we demonstrate the results on the datasets in Table 10 for reference.

### C.2. Experiment on other evaluation metrics

Table 11 demonstrates the results on anomaly prediction evaluated using VUS-PR and VUS-ROC proposed in (Paparrizos et al., 2022). VUS-PR and VUS-ROC are parameter-free measures that are extended from AUC-based measures. As shown in the table, our proposed A2P showed superior performance across various forecasting length, indicating the robustness to predict future anomalies of A2P.

### C.3. Experiment on other baselines

To demonstrate the more generalized performance of A2P, the results of additional anomaly detection baseline experiments are summarized in Table 12. We combined previously utilized time series forecasting models with time series anomaly detection models such as TranAD (Tuli et al., 2022), BeatGAN (Zhou et al., 2019), and DiffusionAD (Zhang et al., 2025). Across these baselines, A2P consistently achieved the highest performance.

### C.4. Results on various tolerance

In the Anomaly Prediction task, it is crucial to detect the exact time steps of anomalies. In this regard, the general Point Adjustment strategy is not fit to the Anomaly Prediction task. Therefore, we define $t$ as the number of time steps to allow errors in anomaly detection outputs before and after each time step, which is used to control the difficulty of the task. We conducted experiments by varying $t$ from 1 to $\infty$, where $\infty$ is equivalent to the existing point adjustment setting. As shown in Figure 8, our proposed A2P outperformed on all $t$, implying that A2P can be used in diverse scenarios, from situations where strict localization of time step is required to more relaxed scenarios. In our experiments, we used $t = 50$ as our default setting.

Table 12: Anomaly Prediction results on other baselines using additional anomaly detection models that are excluded in the main paper with $L_{in} = L_{out} = 100$, averaged over 3 random seeds. The **best** and second-best results are highlighted.

| Model | | Dataset | | | | Avg. F1 |
|---|---|---|---|---|---|---|
| F | AD | MBA | Exathlon | SMD | WADI | |
| | TranAD | 61.38 | 17.35 | 34.67 | 57.12 | 42.63 |
| P-TST | BeatGAN | 64.86 | 16.85 | 33.34 | 57.29 | 43.08 |
| | DiffusionAD | 38.79 | 13.43 | 22.14 | 45.98 | 30.08 |
| | TranAD | 58.10 | 18.48 | 28.97 | 59.36 | 41.22 |
| MICN | BeatGAN | 65.15 | 17.29 | 34.84 | 62.88 | 45.04 |
| | DiffusionAD | 36.14 | 15.85 | 25.90 | 46.28 | 31.04 |
| | TranAD | 60.65 | 17.97 | 33.12 | 61.89 | 43.40 |
| GPT2 | BeatGAN | 50.82 | 17.27 | 30.45 | 55.07 | 38.40 |
| | DiffusionAD | 29.74 | 13.76 | 23.88 | 52.21 | 29.89 |
| | TranAD | 62.05 | 17.54 | 33.28 | 63.04 | 43.97 |
| iTransformer | BeatGAN | 63.97 | 17.35 | 32.19 | 59.68 | 43.29 |
| | DiffusionAD | 37.60 | 11.97 | 23.43 | 48.80 | 30.45 |
| | TranAD | 61.80 | 18.15 | 31.73 | 58.60 | 42.57 |
| FITS | BeatGAN | 61.16 | 17.32 | 28.77 | 53.46 | 40.17 |
| | DiffusionAD | 27.79 | 11.96 | 21.68 | 46.95 | 27.09 |
| **A2P (Ours)** | | **67.55** | **18.64** | **36.29** | **64.91** | **46.84** |

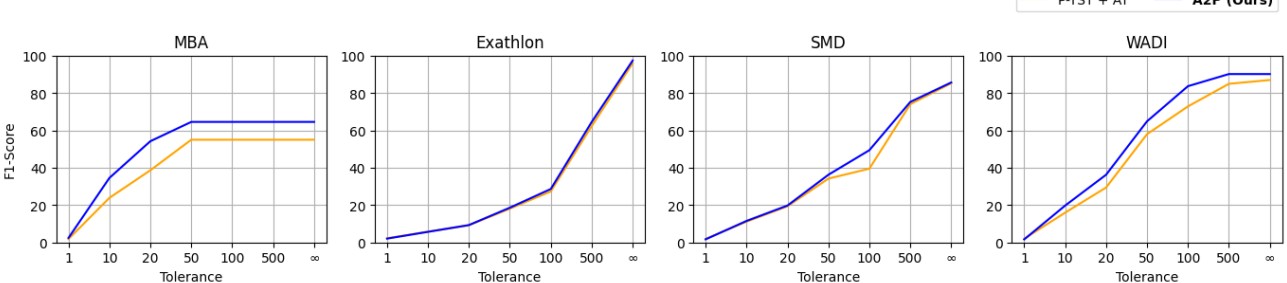

Figure 8: The F1-score of anomaly prediction in various tolerance, when $L_{in} = L_{out} = 100$.

## D. Qualitative Results

We provide additional results of forecasting on the state-of-the-art forecasting models in Figure 9. The result showed that the existing forecasting models fail at predicting abnormal events, since they consider to learn with only normal signals.

## E. Hyperparameter Sensitivity

To figure out the effect of various hyperparameters used in A2P, we examined the F1-scores by varying each hyperparameter, as shown in Figure 10. We conducted experiments with various $\lambda$ coefficients, ranging from 0.1 to 0.9, to weigh each loss term in the objective function. Our proposed model A2P showed stable performance across various values of $\lambda$. Regarding the hyperparameters of Synthetic Anomaly Prompting, we examined the effect of various values of $N$ for the number of anomaly pool, $M$ as the pool size of the anomaly prompt pool, and $L_z$ which is the length of an anomaly prompt. While our proposed A2P achieved stable performance for $N$ and $L_z$ across datasets, $M$ affects the performance. Specifically, the F1-score on the MBA dataset degrades with a bigger pool size, indicating that the selection of an appropriate pool size considering the size of the dataset is needed to fully leverage the effectiveness of SAP. We also examined the influence of $nh_{AFFN}$ which is the number of heads in Anomaly-Aware Forecasting Network. As shown in the last plot of Figure 10, our proposed A2P performed robustly.

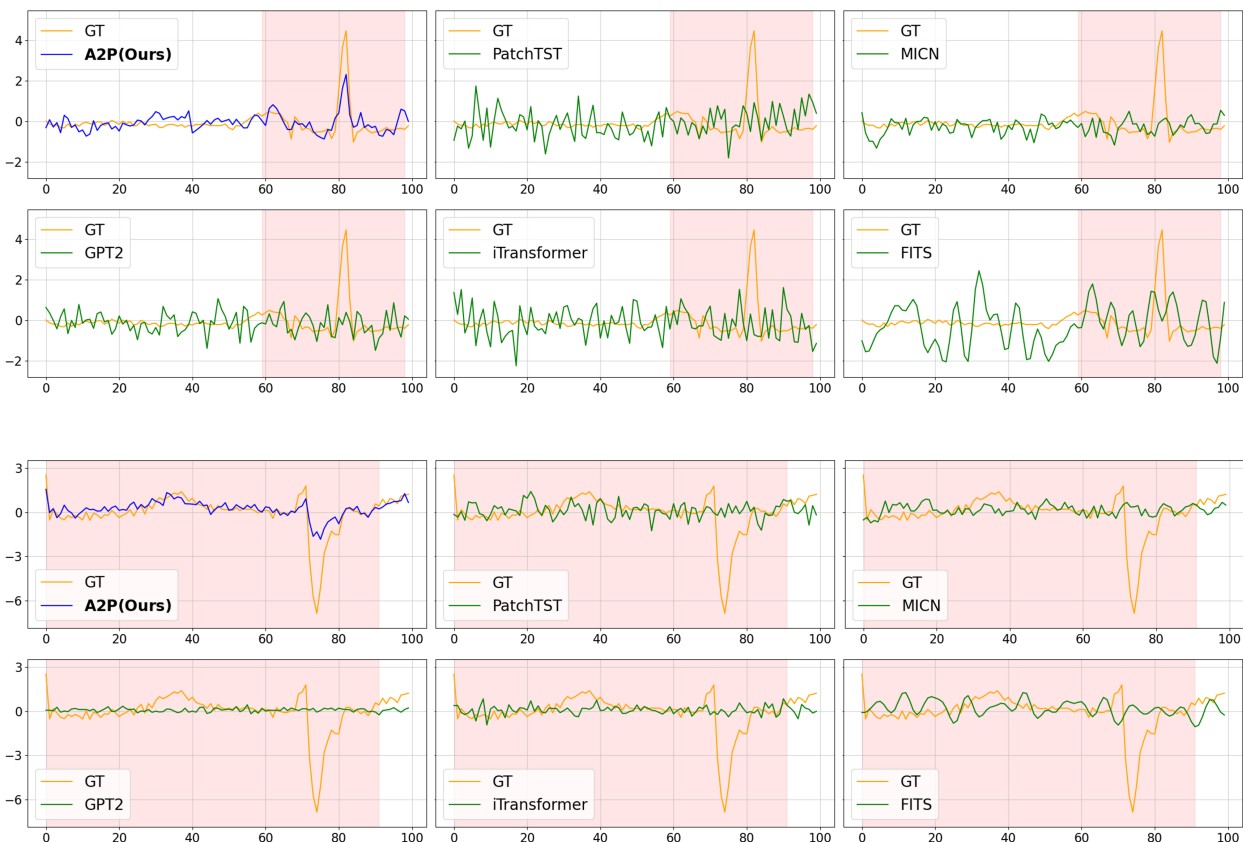

Figure 9: Comparison of the result of forecasting the MBA dataset when $L_{in}$ and $L_{out}$ are 100. Red-shaded area indicate time steps with anomalies.

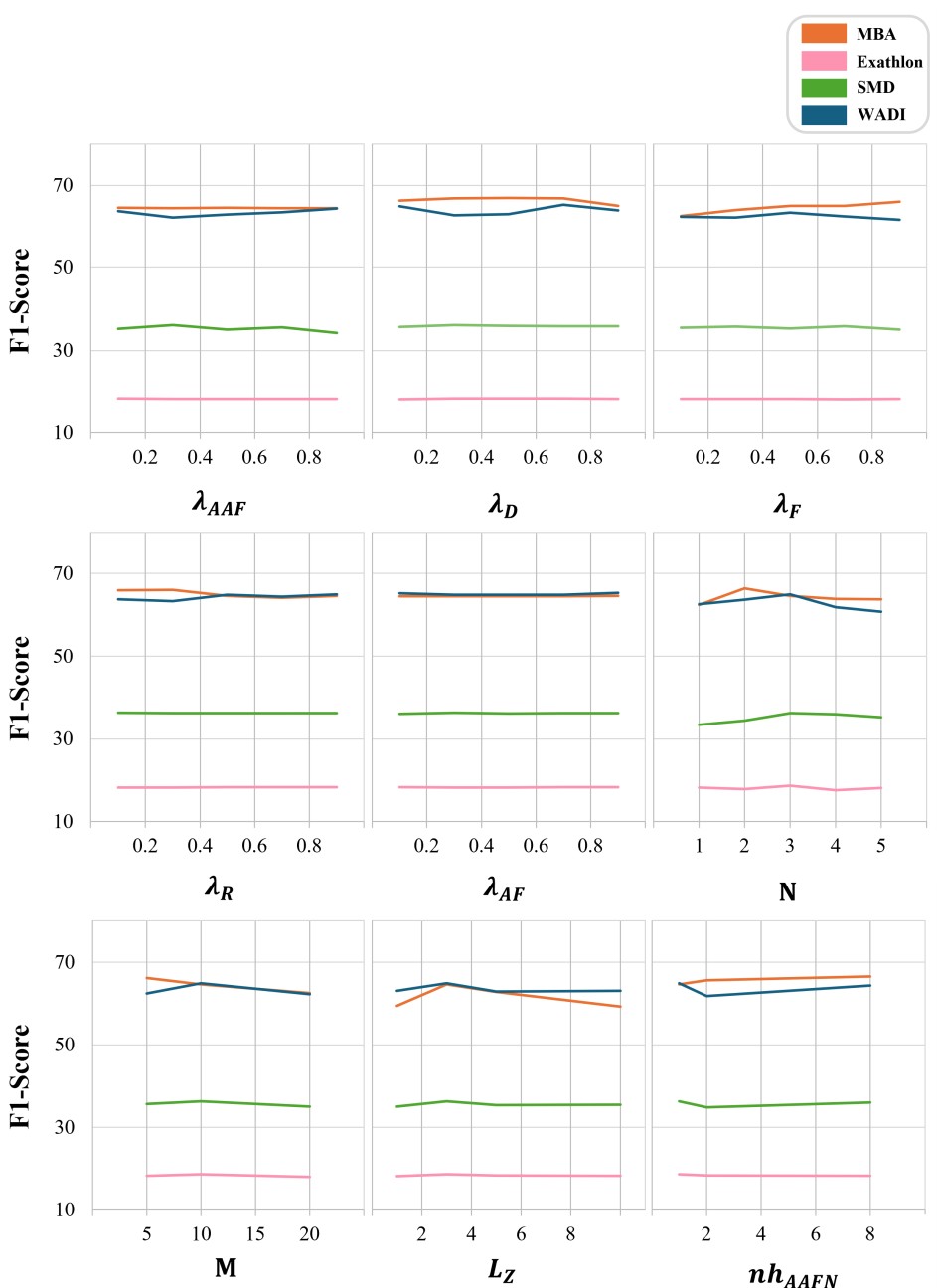

Figure 10: The results on various hyperparameter values when $L_{in} = L_{out} = 100$.

