# OpenReview forum: "When Will It Fail?: Anomaly to Prompt for Forecasting Future Anomalies in Time Series"
_ICML.cc/2025/Conference — ICML 2025 poster_

### Official Review · Reviewer_Xvbm · 2025-02-21

**Overall Recommendation:** 3

**Summary:**

This paper formulates the abnormal prediction problem in time series, aiming to forecast specific future time points where anomalies will occur. Accordingly, the authors propose Anomaly-Aware Forecasting and Synthetic Anomaly Prompting to address the problem.

## update after rebuttal
The authors adequately addressed my concerns, so I raised the score to 3.

**Claims And Evidence:**

I think the results mostly support the claim.

**Essential References Not Discussed:**

Yes.

**Experimental Designs Or Analyses:**

Yes, I checked. The numerical results do not seem very high to me. I am not sure if it is caused by the nature of the task, but anyway, the overall gain looks very substantial.

I am curious about how the authors split the datasets. Do they use non-overlapping windows or overlapping windows to generate samples?

I am also curious about the average anomalies for each sample. As shown in Fig.8, it seems only one animal pattern per sample.

**Methods And Evaluation Criteria:**

Yes. The paper employs some commonly used datasets, but more datasets and more baselines are expected.

Dataset:
- TimeSeAD: Benchmarking Deep Multivariate Time-Series Anomaly Detection

Baselines:
- TranAD: deep transformer networks for anomaly detection in multivariate time series data
- BeatGAN
- TimeMixer
- DiffusionAD:  Imputation-based time-series anomaly detection with conditional weight incremental diffusion models

**Other Comments Or Suggestions:**

N/A

**Other Strengths And Weaknesses:**

The Synthetic Anomaly Prompting seems like a retrieve-based approach for anomaly detection, while integrating it in the framework looks novel to me.

My main concern is that the problem is very naive in real-world applications, and it can be very common in trajectory modeling in automount driving. The cross-attention design is also very intuitive, as one input is a query, and one is a key/value.

Another limitation is that there are too many loss terms and hyper-parameters, making tuning and optimization more complex.

**Questions For Authors:**

- How does the method determine the threshold for the abnormal pattern?
- Can you clarify at which stage which parameters are trainable? Are AFFN trainable at the second stage?  I feel like all components can be trained simultaneously. If this is not optimal, what is the potential reason for that?
- What is [cls] token used for? Is there any specific reason for employing it?

**Relation To Broader Scientific Literature:**

The paper builds upon prior works in time series forecasting and anomaly detection.

**Theoretical Claims:**

There are no theoretical claims. It would be better to include some but not mandatory to me.

---

> ### Author Rebuttal · Authors · 2025-04-01
>
> Thank you for giving us meaningful feedback!
>
> **More Datasets and Baselines**
>
> - TimeSeAD
> |Model|Exathlon-Avg.F1|SMD-Avg.F1|
> |:-:|:-:|:-:|
> |P-TST+AT|14.76|30.58|
> |**A2P(Ours)**|**15.28**|**39.72**|
>
> - TranAD, BeatGAN, TimeMixer, DiffusionAD
> |F model|AD model|Avg.F1|
> |:-:|:-:|:-:|
> |P-TST|TranAD|42.63|
> ||BeatGAN|43.08|
> ||DiffusionAD|30.08|
> |MICN|TranAD|41.22|
> ||BeatGAN|45.04|
> ||DiffusionAD|31.04|
> |GPT2|TranAD|43.40|
> ||BeatGAN|38.40|
> ||DiffusionAD|29.89|
> |iTransformer|TranAD|43.97|
> ||BeatGAN|43.29|
> ||DiffusionAD|30.45|
> |FITS|TranAD|42.57|
> ||BeatGAN|40.17|
> ||DiffusionAD|27.09|
> |TimeMixer|AT|42.72|
> ||DC|34.40|
> ||CAD|17.16|
> ||TranAD|41.85|
> ||BeatGAN|43.55|
> ||DiffusionAD|26.10|
> |**A2P(Ours)**||**46.84**|
>
> Our method outperforms all existing baselines across widely used benchmark datasets, demonstrating its effectiveness and generalizability.
>
> **Numerical Modesty of Results**
>
> AP is challenging as it requires predicting rare anomalies. Even small metric gains are meaningful. Table 1 shows our method consistently outperforms baselines, proving its effectiveness.
>
> **Dataset Splitting**
>
> Following prior works [1], we used non-overlapping windows for AP.
>
> [1] Xu et al. “Anomaly transformer: Time series anomaly detection with association discrepancy.” ICLR 2022.
>
> **Dataset Statistics**
>
> |Dataset|Avg.AnomalyRatio(%)|Avg.AnomalyLen|#AnomalySeg|#Batches|AnomalySeg/TotalSample|AnomalySeg/AnomalySample|
> |:-:|:-:|:-:|:-:|:-:|:-:|:-:|
> |MBA|33.80|29.48|86|75|1.14|1.18|
> |Exathlon|12.69|91.94|1091|8911|0.12|1.00|
> |SMD|4.15|47.26|623|7083|0.08|1.00|
> |WADI|4.88|62.19|27|344|0.07|1.04|
>
> The visualized sample in Fig. 8 typically contains a single anomaly pattern, but our method also handles subtle anomalies effectively, as shown in the table.
>
> **Trajectory Modeling**
>
> AP focuses on forecasting rare anomalies, while trajectory modeling predicts paths under normal conditions [1,2]. Some works [3] focus on anomaly detection, not future predictions, so AP remains underexplored in this context. A2P’s core idea could be explored in trajectory modeling in future research.
>
> [1] Tang et al. “HPNet: Dynamic Trajectory Forecasting with Historical Prediction Attention.” CVPR 2024.
>
> [2] Phan-Minh et al. “CoverNet: Multimodal Behavior Prediction using Trajectory Sets.” CVPR 2020.
>
> [3] D'amicantonio et al. “uTRAND: Unsupervised Anomaly Detection in Traffic Trajectories.” CVPR 2024.
>
> **Cross-attention**
>
> While the cross-attention design may seem intuitive, the core contribution of AAF lies not in the architecture itself but in the idea of learning the inherent patterns of prior and posterior signals to predict future anomalies. By modeling the relationship between these signals, AAF captures temporal dependencies that are critical for AP, going beyond traditional forecasting or anomaly detection approaches.
>
> **Loss terms and Hyperparameters**
>
> Our method introduces two unique loss terms, $L_{AAF}$ and $L_{D}$, with others extended from traditional time series loss functions. A2P shows strong robustness across hyperparameter settings,as shown in Section E of the supplementary material. This suggests that they do not significantly complicate the tuning process.
>
> **Anomaly Threshold**
>
> The anomaly threshold follows the widely accepted protocol from [1], adjusting for a percentage of anomalies in the test data. This approach ensures consistency with established standards for anomaly detection tasks.
>
> [1] Shen et al. "Timeseries anomaly detection using temporal hierarchical one-class network." NeurIPS 2020.
>
> **Trainable Parameters**
>
> In the pre-training stage, only AAFN and APP parameters are trainable. In the main training phase, only the backbone parameters are trained.
>
> **Simultaneous Training**
>
> |Method|MBA|Exathlon|SMD|WADI|Avg.F1|
> |:-:|:-:|:-:|:-:|:-:|:-:|
> |Simultaneous Training|45.70|17.90|33.26|59.86|39.18|
> |**Ours**|**67.55**|**18.64**|**36.29**|**64.91**|**46.84**|
>
> We experimented with training all components simultaneously and found it suboptimal as shown in the table. This may be due to the Anomaly Probability output from the Anomaly-Aware Forecasting Network, which, when not fully trained, hinders proper forecasting. By using a two-stage training strategy, we ensure that the anomaly probability is learned first, allowing it to enhance future time series prediction during the main training phase, leading to better performance.
>
> **[CLS] Token**
>
> The [cls] token is a learnable embedding used to capture global representations, similar to its role in BERT [1]. It helps select the most relevant anomaly prompt in A2P, enabling effective abnormal signal synthesis.
>
> [1] Devlin et al. "Bert: Pre-training of deep bidirectional transformers for language understanding." NAACL 2019.
>
>
> We greatly appreciate your helpful comments, and we will be sure to include the above-mentioned details in our revision. This will provide a more thorough explanation and address the points you raised in a comprehensive manner, enhancing the overall quality of the paper.

---

> > ### Comment · Reviewer_Xvbm · 2025-04-06
> >
> > Sorry for the late response. I thank the authors for addressing most of the concerns. I decided to raise my score, and the paper can definitely be accepted.
> >
> > I still think predicting future anomalous or risky scenarios has been explored in fields such as autonomous driving and motion planning, where the aim is to anticipate and avoid potential hazards or obstacles (these tasks are a kind of spatial-temporal analysis, which is related to time series analysis).
> > I suggest authors can include more discussion in their final version. **Given the limited time, no response here is totally fine with me :)**

---

> > > ### Author Response · Authors · 2025-04-08
> > >
> > > Thank you for the thoughtful follow-up and for raising your score. While we acknowledge related efforts in autonomous driving and motion planning, we would like to clarify key differences between those and our proposed Anomaly Prediction (AP) task. The closest setting in autonomous driving is early accident anticipation [1, 2], which focuses on detecting the possibility of an accident as early as possible within a short video clip, without predicting when it will happen. In contrast, our AP task aims to forecast both **if** and **when** an anomaly will occur, often requiring longer lead times and carefully consider subtle signals—making it more general and more challenging. We believe these aspects highlight the novelty and difficulty of AP, setting it apart from the related work. Thank you again for your valuable feedback, and we will incorporate this discussion into our revised paper.
> > >
> > > [1] When, Where, and What? A Novel Benchmark for Accident Anticipation and Localization with Large Language Models (ACM MM 2024)
> > >
> > > [2] Graph(Graph): A Nested Graph-Based Framework for Early Accident Anticipation (WACV 2024)

---

### Official Review · Reviewer_ADJk · 2025-02-28

**Overall Recommendation:** 4

**Summary:**

This paper proposes a novel framework, Anomaly to Prompt (A2P), to address the Anomaly Prediction (AP) task in time series analysis, which aims to predict future anomalies. The framework integrates two key components: Anomaly-Aware Forecasting (AAF) that learns relationships between anomalies and future signals, and Synthetic Anomaly Prompting (SAP) generates synthetic anomalies through signal-adaptive prompting. Experiments on real-world datasets demonstrate A2P’s superiority over SOTA methods.

**Claims And Evidence:**

Yes.

**Essential References Not Discussed:**

No. Related works are well discussed.

**Experimental Designs Or Analyses:**

Yes, the experimental designs are reasonable.

**Methods And Evaluation Criteria:**

Yes, the proposed method and evaluation criteria makes sense.

**Other Comments Or Suggestions:**

Please refer to the weakness.

**Other Strengths And Weaknesses:**

Strengths:
1. The paper fills a critical gap between traditional anomaly detection (AD) and forecasting by defining Anomaly Prediction (AP) as a distinct task requiring precise localization of future anomalies.
2. The proposed A2P achieves state-of-the-art F1 scores across different datasets.
3. The proposed model is easy to follow and codes are provided.

Weaknesses:
1. The initialization and optimization details of the the proposed APP are unclear. For instance, how are prompts initialized?
2. Computational costs for pre-training AAF and APP are not quantified, which is critical for real-world deployment.

**Questions For Authors:**

1. Since the anomaly prediction (AP) task fundamentally differs from anomaly detection (AD) methods, the authors construct baselines with combinations of forecasting and anomaly detection models. Could the proposed model be compared with existing models that were specifically designed for AP tasks?
2. Can the model detect precursors of impending anomalies? For example, before an anomaly occurs (the status of the device is still normal), the model can identify early warning signals in the data, enabling interpretable anomaly prediction.

**Relation To Broader Scientific Literature:**

The key contributions of this paper bridges the gap between time series forecasting and anomaly detection, which provides insights for future study in anomaly prediction rather than the popular anomaly detection tasks.

**Theoretical Claims:**

I think the theoretical claims of this paper are correct.

---

> ### Author Rebuttal · Authors · 2025-04-01
>
> We are sincerely grateful for giving us positive comments and acknowledging the contribution of our proposed method A2P for tackling the challenges of AP.
>
> **Initialization and Optimization Details**
>
> All additional parameters introduced for APP are initialized using a standard uniform initialization method in the PyTorch library. We will include this detail in the revision.
>
> **Computational Complexity**
>
> - To investigate the impact of additional computational complexity of our proposed methods, we measured GFLOPs per iteration (including both pre-training and main training) and the total number of parameters of the baseline (combination of PatchTST and AnomalyTransformer) and our proposed A2P.
> As shown in the figure (URL:https://ifh.cc/v-SRor7L), A2P does require additional computation compared to the baseline. However, since our model unifies both forecasting and anomaly detection within a single framework, it significantly reduces the overall parameter footprint. More importantly, this additional cost is only incurred during training, with no extra overhead at inference time. Despite the modest increase in training complexity, A2P achieved an average 10% improvement in performance over the baseline, demonstrating a favorable trade-off between cost and accuracy. Moreover, the train time consumed is utmost 1 hour in WADI dataset, which is negligible and is not really a heavy burden for A2P to be applied in real-world scenarios.
> - Regarding scalability, the four datasets in Table 1 cover various ranges of dataset scales, as well as dynamic environments. To further validate A2P’s effectiveness, we conducted additional experiments on the KPI dataset (representing large-scale data) and the NeurIPS-TS dataset (characterized by highly dynamic anomaly patterns across all five anomaly types specified in [1]). A2P achieved state-of-the-art F1-score on both datasets, highlighting its robustness and scalability in large and dynamic settings.
>
> |Model|NeurIPS-TS|KPI|
> |:-:|:-:|:-:|
> |P-TST+AT|23.09|24.49|
> |**A2P(Ours)**|**34.18**|**33.46**|
>
> [1] Lai et al. "Revisiting time series outlier detection: Definitions and benchmarks." NeurIPS 2021.
>
> **Existing Method for AP**
>
> To the best of our knowledge, there are currently no existing methods specifically designed for the AP task. However, a related study [1] conducted AP experiments using existing forecasting models, though these approaches do not directly address the unique challenges of AP.
> For a fair comparison, we constructed baselines by systematically combining established forecasting and anomaly detection models. This setup allows us to benchmark our method against the most relevant alternatives and highlights its distinct advantages.
>
> [1] You et al. "Anomaly Prediction: A Novel Approach with Explicit Delay and Horizon." ICCP 2024.
>
> **Precursor Detection**
>
> Our proposed model A2P can detect precursors of impending anomalies. Since there are no labels indicating ‘precursor of anomalies’ in the dataset, we cannot measure the numeric performance of detection of precursors explicitly.
> Instead, we visualized the attention map of the cross attention in Anomaly-Aware Forecasting Network, to investigate if the model can relate specific parts of the prior signals when identifying future anomalies, as shown in the figure (URL:https://ifh.cc/v-cQD6Ka). When identifying the future anomalies (red-shaded area), which is the main purpose of AP, the model focuses on specific area of prior signals (circled area). While not perfect, our proposed model can provide a further explainable information to infer which prior time steps contribute to predicting future anomalies. This approach can be effective in many real-world scenarios, for example, medical doctors can scrutinize the normal data of patients to give them appropriate instructions to prevent possible diseases.
>
>
> We will incorporate the additional details such as initialization and computational costs in the revision, and thank you again for your careful attention.

---

> > ### Comment · Reviewer_ADJk · 2025-04-02
> >
> > The authors have solved my problems. In my opinion, anomaly prediction is a more interesting and meaningful task than traditional anomaly detection, and the authors have provided a solution to it. Thus, I have raised my rating.

---

> > > ### Author Response · Authors · 2025-04-02
> > >
> > > We truly appreciate your kind reassessment and glad that your concerns have been resolved. Thank you also for your thoughtful review and the time you devoted to evaluating our manuscript. Should any further questions arise, we would be grateful for your continued feedback.

---

### Official Review · Reviewer_K8J2 · 2025-03-12

**Overall Recommendation:** 3

**Summary:**

The paper introduces a novel framework called A2P designed to forecast future anomalies in time series data. Unlike traditional forecasting models—typically trained on standard signals and consequently fail to accurately predict abnormal events—the proposed method integrates anomaly-aware components into the forecasting process. A2P is comprised of two key elements:

- **Anomaly-Aware Forecasting (AAF):** This component pre-trains a forecasting network to learn the relationships between past anomalies and future trends, enabling the model to predict signals more accurately and reflect potential abnormal events.
- **Synthetic Anomaly Prompting (SAP):** This technique employs a learnable Anomaly Prompt Pool (APP) to inject synthetic anomalies at the embedding level during training. This process helps the model simulate and recognize various anomaly patterns.

The authors validate their approach with extensive experiments on multiple real-world datasets, demonstrating significant improvements in anomaly prediction and forecasting accuracy over existing baselines.

**Claims And Evidence:**

**Evidence Supporting the Claims:**
The paper provides extensive empirical evaluations on several real-world datasets, including MBA, Exathlon, SMD, and WADI, which strongly support the core claims. The experimental results, including detailed ablation studies, convincingly demonstrate that the proposed A2P framework—through its components of Anomaly-Aware Forecasting (AAF) and Synthetic Anomaly Prompting (SAP)—achieves significant improvements in both forecasting accuracy and anomaly prediction performance compared to established baselines.

**Problematic Aspects:**
 The integration of AAF and SAP introduces additional computational complexity. The paper does not provide a detailed evaluation of the computational trade-offs or scalability of the proposed approach in large-scale or highly dynamic environments.

In addition, given that similar work had been done before, the author claims that “we first propose a method to deal with the problems
of Anomaly Prediction.” is suspect.

**Essential References Not Discussed:**

The submission provides a solid set of citations for time series forecasting and anomaly detection, including recent works on anomaly prediction. However, some additional related works could further contextualize its key contributions:

- **Prompt-Based Learning Literature:**
  The Synthetic Anomaly Prompting component is reminiscent of prompt tuning techniques that have seen significant development in the NLP community [1]. While the paper adapts the concept to time series data, discussing these foundational works could help readers understand the conceptual lineage and justify the design of the learnable Anomaly Prompt Pool.

- **Synthetic Data Generation for Anomalies:**
  Although the authors reference anomaly injection schemes (e.g., Darban et al., 2025), other works focus on synthetic anomaly generation and data augmentation in time series that might offer additional insights or alternative approaches. A broader review of such methods could strengthen the rationale for the proposed synthetic anomaly prompting strategy.

- **Unified Architectures for Forecasting and Anomaly Detection:**
  The paper contributes by merging forecasting and anomaly detection within a shared architecture. There is an emerging literature on integrated approaches in this space. While the submission cites several relevant studies, additional references that discuss unified or multi-task learning frameworks for time series analysis could further clarify how the proposed method fits into and extends current trends.

[1]Brian Lester, Rami Al-Rfou, and Noah Constant. 2021. The Power of Scale for Parameter-Efficient Prompt Tuning. In Proceedings of the 2021 Conference on Empirical Methods in Natural Language Processing, pages 3045–3059, Online and Punta Cana, Dominican Republic.
Association for Computational Linguistics.

**Experimental Designs Or Analyses:**

**Experimental Design Validity and Analysis:**
I reviewed the experimental setups detailed in the submission, which include evaluations of both forecasting and anomaly prediction capabilities. The experiments are conducted on multiple real-world datasets (MBA, Exathlon, SMD, and WADI), ensuring diverse and representative scenarios. The following points summarize my assessment:

- **Soundness of Evaluation Metrics:**
  The use of Mean Squared Error for assessing forecasting accuracy and F1-score (with a tolerance window for anomaly detection) is appropriate and well-justified for the tasks at hand.

- **Ablation Studies:**
  Comprehensive ablation experiments are presented to isolate the contributions of the two key components: Anomaly-Aware Forecasting (AAF) and Synthetic Anomaly Prompting (SAP). These studies effectively demonstrate the individual and combined benefits of each component.

**Methods And Evaluation Criteria:**

The proposed methods are well-aligned with anomaly prediction in time series data. Integrating Anomaly-Aware Forecasting and Synthetic Anomaly Prompting directly addresses the challenge of forecasting future anomalies—a problem where traditional forecasting methods often fall short. The experimental evaluation, which uses benchmark datasets such as MBA, Exathlon, SMD, and WADI, is appropriate for the application, as these datasets represent diverse real-world scenarios in domains like medical monitoring and industrial systems. Additionally, relevant metrics (e.g., F1-score with tolerance for anomaly detection and Mean Squared Error for forecasting) provide clear quantitative evidence of the framework's performance. The problem is that the author's description of his newly created evaluation index is too brief for the controversial F1-related evaluation criteria. More detailed explanations should be given for the evaluation indicators to avoid confusion among readers.

**Other Comments Or Suggestions:**

**Other Comments and Suggestions:**
Overall, the paper is well-structured and presents its ideas in a technically detailed manner. Nonetheless, a few minor points could further enhance its clarity and presentation:

- **Typographical and Formatting Consistency:**
  Although a search for explicit typographical errors did not reveal any major issues, I observed occasional minor inconsistencies in formatting. For example, there are instances where spacing around mathematical symbols and figure captions could be more uniform. Additionally, the notational switch between “AAF” and “AAFN” (when referring to the anomaly-aware forecasting components) might confuse readers; ensuring consistency in the notation throughout the manuscript would improve readability.

- **Clarity in Complex Sections:**
  The technical sections are dense, particularly those describing the Synthetic Anomaly Prompting (SAP) module and its associated loss functions. Consider incorporating additional intuitive explanations or visual aids to help demystify these complex components for a broader audience.

- **Proofreading for Minor Errors:**
  A careful proofreading to check for any minor grammatical errors or inconsistencies in punctuation could further polish the manuscript.

**Other Strengths And Weaknesses:**

- **Originality:**
  The paper exhibits originality by creatively combining ideas from time series forecasting, anomaly detection, and prompt-based learning. The introduction of Synthetic Anomaly Prompting (SAP) via a learnable Anomaly Prompt Pool (APP) is a novel adaptation of concepts initially developed in the NLP domain. Additionally, integrating Anomaly-Aware Forecasting (AAF) to bridge the gap between regular signal forecasting and anomaly detection provides a fresh perspective on the long-standing challenge of forecasting anomalies.

- **Significance:**
  Addressing the problem of forecasting future anomalies has considerable practical significance, particularly in applications such as medical monitoring and industrial system maintenance. The proposed framework demonstrates improved predictive performance over existing baselines and has the potential to influence the design of early-warning systems in critical domains. This application-driven impact underscores the broader importance of the contributions.

- **Clarity:**
  The paper is technically rigorous and detailed, clearly describing its methodology and experimental setups. However, the presentation can be dense and highly technical in certain sections, which might impede accessibility for a broader audience. Improving the narrative flow or providing additional intuition behind complex components could enhance clarity without sacrificing depth.

- **Additional Strengths:**
  The extensive empirical evaluations across multiple benchmark datasets and comprehensive ablation studies lend strong support to the proposed approach. These experimental results validate the effectiveness of the individual components (AAF and SAP) and demonstrate the unified architecture's benefits in learning robust representations for forecasting and anomaly detection.

- **Additional Weaknesses:**
No code was provided for verification.

**Questions For Authors:**

1. Can you elaborate on the sensitivity of the Synthetic Anomaly Prompting (SAP) component to hyperparameter settings (e.g., the size of the anomaly prompt pool and the number of prompts attached)?

2. Could you discuss the integrated A2P framework's computational complexity and scalability, especially compared to standard forecasting methods?

3. Are there specific scenarios or types of datasets where the A2P method underperforms relative to existing baselines? If so, can you provide an analysis of these cases and potential strategies for improvement?

4. This work would be more convincing if a reproducible verification code were provided.

**Relation To Broader Scientific Literature:**

The paper’s contributions build on and extend several key ideas from the broader literature on time series analysis, anomaly detection, and forecasting. Specifically:

- **Integration of Forecasting and Anomaly Detection:**
  Traditional time series forecasting methods (e.g., PatchTST, GPT2, iTransformer) are designed to predict normal behavior and, thus, often neglect anomalies. In contrast, this work builds on the observation—also noted in recent studies such as Jhin et al. (2023) and You et al. (2024)—that forecasting models need to account for abnormal signals to be truly effective in real-world scenarios. By embedding anomaly awareness into the forecasting process, the paper extends prior work that typically treats forecasting and anomaly detection as separate tasks.

- **Anomaly-Aware Forecasting (AAF):**
  The idea of pre-training a forecasting network to learn the relationship between past anomalies and future trends is novel in its application to anomaly prediction. While earlier studies have focused on detecting anomalies from historical data or near-term forecasts, this approach leverages anomaly-aware pre-training to predict future abnormal events, thereby addressing a gap in the literature.

- **Synthetic Anomaly Prompting (SAP):**
  Using a learnable Anomaly Prompt Pool (APP) to inject synthetic anomalies during training is conceptually related to techniques in data augmentation and prompt-based learning seen in other areas of machine learning. However, its application in time series anomaly prediction is innovative. This idea builds on earlier work in synthetic data generation and augmentation but adapts these principles to enhance the representation of anomalies, thereby improving detection accuracy in forecasting future signals.

**Theoretical Claims:**

The submission primarily focuses on algorithmic innovations and empirical validation rather than on formal theoretical proofs. While the paper does provide mathematical formulations of its loss functions and training objectives (such as LAAF, LD, and LF), it does not include formal proofs for theoretical claims. I verified that the provided formulations are logically consistent with the intended design of the A2P framework; however, since no formal proofs were presented, there was no need to check the correctness of theoretical proofs in a rigorous sense.

---

> ### Author Rebuttal · Authors · 2025-04-01
>
> We appreciate your meaningful feedback, and your insights are invaluable as we continue to improve our work!
>
> **Computational Complexity**
>
> Please refer to the Computational Complexity section in Reviewer ADJk.
>
> **AP Task**
>
> The only prior work on Anomaly Prediction in time series is [1], which only formulates the problem without proposing concrete solutions. In contrast, our work is the first to introduce practical methods for AP, specifically through SAP and AAF, which directly address the core challenges and advance the field.
>
> [1] You et al. "Anomaly Prediction: A Novel Approach with Explicit Delay and Horizon." ICCP 2024.
>
> **Underspecified Metrics**
>
> The conventional point adjustment for evaluation, as discussed in [1], has known limitations. Since the goal is to predict anomalies within a reasonable time window rather than at exact points, using raw prediction outputs is not ideal. For instance, medical doctors are more concerned with detecting abnormal symptoms over a time window, rather than at specific seconds or minutes. To address this, we use the F1-score with tolerance, a modified version of the traditional F1 with point adjustment, which allows error tolerance with a set time window around predicted anomalies, offering a more realistic evaluation, as shown in the figure (URL: https://ifh.cc/v-ZLV6Kf).
>
> [1] Kim et al. "Towards a rigorous evaluation of time-series anomaly detection." AAAI 2022.
>
> **Data Augmentation**
>
> ||NoAug|Spikes|Context|Flip|Noise|CutOff|Scale|Wander|Avg|**A2P(Ours)**|
> |-|:-:|:-:|:-:|:-:|:-:|:-:|:-:|:-:|:-:|:-:|
> |Avg. across datasets (F1)|38.89|39.18|37.62|39.27|37.60|37.63|39.56|38.94|38.76|**46.84**|
>
> We compared A2P with data augmentation methods from [1]. While these methods improve prediction to some extent, they are limited by fixed anomaly injection rules. A2P, with context-aware prompts, generates more realistic anomalies, yielding better performance.
>
> [1] Goswami et al. “Unsupervised model selection for time series anomaly detection.” ICLR 2023.
>
> **Unified Framework**
>
> Additional references on unified and multi-task frameworks, like [1] and [2], will be discussed. However, these models are unsuitable for AP. [2] only focuses on forecasting and simply combining forecasting and anomaly detection modules-such as in foundation models that handle each task independently-performs poorly, as shown in Table 1 baselines. AP requires forecasting anomalies and learning temporal patterns, which existing models lack. In contrast, A2P is specifically built for AP, integrating AAF with a learnable prompt pool to extend unified modeling trends into proactive anomaly handling.
>
> [1] Gao et al. “UniTS: A unified multi-task time series model.” NeurIPS 2024.
>
> [2] Woo et al. “Unified training of universal time series forecasting transformers.” ICML 2024.
>
> **Method Flow**
>
> Our two methods—SAP and AAF—work together in the A2P framework to bridge anomaly detection and forecasting. SAP addresses the challenge of limited abnormal signals in training data by using trainable anomaly prompts to create realistic synthetic anomalies, enhancing the model's anomaly recognition. AAF, in contrast, forecasts anomalies directly, improving predictions in abnormal conditions. Together, SAP and AAF complement each other—SAP enriches training data, while AAF uses this enriched data to enhance forecasting signals with anomalies. This synergy allows A2P to effectively detect and predict anomalies.
>
> **SAP Explanation**
>
> To clarify the SAP module, we include a figure (URL: https://ifh.cc/v-3dyYJd) that illustrates its mechanism. The Anomaly Prompt transforms normal features into anomalies, and the loss function $L_D$ guides the prompt pool to generate meaningful anomalies.
>
> **Grammatical Errors**
>
> We will carefully proofread the manuscript to correct any minor grammatical errors, e.g., “we formulate called Anomaly Prediction” or “are jointly pre-train”, and inconsistencies in punctuation, ensuring a more polished and professional presentation.
>
> **Hyperparameter Sensitivity**
>
> Please refer to the Loss terms and Hyperparameters section in Reviewer Xvbm.
>
> **Extra Anomaly Scenarios**
>
> |Model|Point Global|Point Context|Contextual Global|Contextual Seasonal|Contextual Trend|Avg|
> |:-:|:-:|:-:|:-:|:-:|:-:|:-:|
> |P-TST+AT|2.79|3.04|54.59|59.28|55.80|35.10|
> |**A2P(Ours)**|**8.29**|**8.97**|**75.65**|**72.32**|**61.93**|**45.43**|
>
> We evaluated A2P on five univariate NeurIPS-TS datasets, covering various anomaly types, as outlined in [1]. A2P outperformed the baseline in F1-score, especially on contextual anomalies. However, it struggles with point anomalies due to their short duration. Future work could explore adding context or using multi-step forecasting to better detect point anomalies.
>
> [1] Lai et al. "Revisiting time series outlier detection: Definitions and benchmarks." NeurIPS 2021.
>
> **Code Provision**
>
> We provide the reproducible code in anonymous repository https://anonymous.4open.science/r/A2P-E2FC.

---

> > ### Comment · Reviewer_K8J2 · 2025-04-04
> >
> > The author's reply, to some extent, answers my doubts. Additionally, something that needs to be considered is the description of evaluation indicators on page 6, lines 281-287: "In addition, Fl-score was calculated without point adjustment introduced in (Audibert et al., 2020). Instead, we used F1-score with tolerance t ...". It is widely known that the point adjustment evaluation method is already a very lenient and easy-to-exaggerate indicator of the actual performance of the model, and the description in this work seems to adopt an even more lenient evaluation indicator. Therefore, the practicality and accuracy contribution of this work may be questioned. For this point, I hope to receive further detailed explanation and clarification.

---

> > > ### Author Response · Authors · 2025-04-05
> > >
> > > Thank you for the valuable feedback and the opportunity to further clarify our evaluation strategy. As you pointed out, the original point adjustment evaluation method can exaggerate the performance of anomaly detection, which is a well-known issue in the field. To address this, in our evaluation, we adopted the F1-score with tolerance $t$, which uses **a fixed time window** ($±t$) around each predicted anomaly point, rather than considering the entire anomaly segment window – the main limitation of the point-adjusted F1-score (please refer to the following figure: https://anonymous.4open.science/r/A2P-E2FC/png/tolerance.png).
> > >
> > > **Therefore, we would like to emphasize that the F1-score with tolerance $t$ is an even stricter, more application-aligned, and practical evaluation metric compared to the point-adjusted F1-score, rather than being a more lenient metric.**
> > >
> > > We will revise the paper to clearly describe the motivation and advantages of our evaluation metric, with a more detailed explanation.
> > > We thank you again for your thoughtful feedback, and please do not hesitate to reach out with any further questions regarding the evaluation metric.

---

### Decision · Program_Chairs · 2025-05-01

**Decision:**

Accept (poster)

**Comment:**

This paper tackles the task of Anomaly Prediction in time series, proposing A2P that integrates anomaly-aware forecasting with synthetic prompt–based training.

Reviewers likes it for address a gap that conventional forecasting and detection methods overlook, since they rarely predict future anomalous time points. The authors’ experiments on real datasets show their method outperforms combined baselines and earlier approaches. Ablations are also good!

 Some noted that the tolerance-based F1 metric needs careful interpretation, but the authors clarified it is stricter than some existing point-adjusted scores. Additional comparisons (e.g., with time-series baseline) further strengthen the claim.

In sum, we find A2P to be a well-motivated solution to anomaly prediction with solid experimental gains, recommending acceptance.